# Finger sweat analysis enables short interval metabolic biomonitoring in humans

Julia Brunmair [1,4], Mathias Gotsmy [1,4], Laura Niederstaetter[1], Benjamin Neuditschko [1,2], Andrea Bileck [1,3], Astrid Slany [1], Max Lennart Feuerstein[1], Clemens Langbauer[1], Lukas Janker [1,3], Jürgen Zanghellini [1], Samuel M. Meier-Menches [1,2,3] & Christopher Gerner [1,3✉]

Metabolic biomonitoring in humans is typically based on the sampling of blood, plasma or urine. Although established in the clinical routine, these sampling procedures are often associated with a variety of compliance issues, which are impeding time-course studies. Here, we show that the metabolic profiling of the minute amounts of sweat sampled from fingertips addresses this challenge. Sweat sampling from fingertips is non-invasive, robust and can be accomplished repeatedly by untrained personnel. The sweat matrix represents a rich source for metabolic phenotyping. We confirm the feasibility of short interval sampling of sweat from the fingertips in time-course studies involving the consumption of coffee or the ingestion of a caffeine capsule after a fasting interval, in which we successfully monitor all known caffeine metabolites as well as endogenous metabolic responses. Fluctuations in the rate of sweat production are accounted for by mathematical modelling to reveal individual rates of caffeine uptake, metabolism and clearance. To conclude, metabotyping using sweat from fingertips combined with mathematical network modelling shows promise for broad applications in precision medicine by enabling the assessment of dynamic metabolic patterns, which may overcome the limitations of purely compositional biomarkers.

[1] Department of Analytical Chemistry, Faculty of Chemistry, University of Vienna, Vienna, Austria. [2] Department of Inorganic Chemistry, Faculty of Chemistry, University of Vienna, Vienna, Austria. [3] Joint Metabolome Facility, University and Medical University of Vienna, Vienna, Austria. [4]These authors contributed equally: Julia Brunmair, Mathias Gotsmy. ✉email: christopher.gerner@univie.ac.at

Metabolic phenotyping seeks to identify biomarkers for diagnosis, prognosis or therapy and holds great promise to improve clinical practice and especially, precision medicine[1,2]. Despite considerable progress with respect to the sensitive and parallel analysis of metabolites in metabolomics/ metabonomics studies[3–7] and by mass spectrometry (MS)[8,9], the successful implementation of metabolites as biomarkers in the clinical setting still represents a major challenge[10–12]. This is illustrated by the strong individual and physiological background variability[2] and individual differences in ADME properties, the latter impacting significantly on drug responses[13,14]. To the best of our knowledge, current techniques of metabolic phenotyping are largely focussed on generating static diagnostic pictures because the commonly used biological fluids (e.g. plasma, urine)[15–17] or tissues do not routinely allow for time-course studies. The implementation of dynamic metabolic responses as a biomarker strategy may be desirable, but requires a considerable number of data points on a single individual. Clearly, a non-invasive method from an alternative biological fluid is required to enable frequent sampling of the same individual in order to obtain dynamic metabolic patterns in the frame of metabolic phenotyping.

While fingerprints—the pattern of the ridge details left on a surface—have been used for the identification of individuals since the late 19th century[18], their relevance for detecting metabolites, as well as drugs and their metabolites has only recently been discovered[19,20]. While drug substances detected in the fingerprint may originate from accidental dermal contact, the detection of drug-specific metabolites implies that the drug was ingested, metabolised and subsequently excreted from sweat glands. Thus, we hypothesised that sweat from the skin surface may represent a promising source for metabolic biomonitoring. Sweat is a hypotonic, slightly acidic biofluid secreted by the eccrine, apocrine and apoeccrine glands located on the skin surface[21,22]. Eccrine sweat from the fingertips is mainly composed of water (~99%), but contains electrolytes, urea, lactate, amino acids, metal ions[23,24] and a variety of endogenous metabolites, including peptides, organic acids, carbohydrates, lipids, lipid-derived metabolites, as well as xenobiotics[21,22,25–27]. Sweat composition is highly dynamic, changes significantly with pathological states and may reveal habits of diet, metabolic conditions or use of drugs and supplements[17,24,28]. In fact, the analysis of sweat has already been reported to assess individual metabolic characteristics[29,30]. Clinical assays based on the analysis of sweat exist and include the screening of newborn children for elevated chloride and sodium levels to confirm cystic fibrosis via pilocarpine stimulated iontophoresis or forensic and criminal investigations to test for illicit drug use[17,22,31–33]. Furthermore, it has already been successfully demonstrated that the analysis of proteins contained in sweat enables not only the diagnosis of active tuberculosis but can also be used to screen for lung cancer[16,34,35], highlighting the potential of sweat analysis for precision medicine[36]. Real-time monitoring of biomarkers was demonstrated with wearable sweat sensors for uric acid and tyrosine[37], interleukin-6 and cortisol[38] or electrolytes such as sodium, ammonium ions and lactate[39].

However, these studies typically assessed a small number of metabolites and relied on elaborate methods to collect sweat, including sweat patches or artificially forcing sweat production[17,22,30]. This was necessary because the detection methods required relatively large absolute amounts of these metabolites. It is known that eccrine glands on the fingertips produce sweat at a rate of 50–500 nL cm$^{-2}$ min$^{-1}$ [40]. Thus, the analysis of metabolites from sweat of the fingertips may be achieved with sufficiently sensitive instrumentation, for example MS[41]. Sample collection using sweat from fingertips requires no patient pre-treatment or trained personnel, is safe and fast. Upon optimising the entire workflow for the analysis of sweat from the fingertips, we analysed 1792 samples from 40 participants, which underlines its potential as a high-throughput metabolic technology. Proof-of-principle studies based on the consumption of coffee or ingestion of a caffeine capsule were designed to assess metabolic time-series of each participant and provided evidence of the feasibility of this approach. Fluctuations in the rate of sweat production were accounted for by mathematical modelling of the conversion of xenobiotics to their catabolic products (e.g. caffeine to paraxanthine). In this study, we show that metabolic phenotyping using sweat from fingertips combined with mathematical network modelling may have far reaching relevance for precision medicine, because it allows to obtain dynamic metabolic responses of individuals.

## Results

**Sweat from the fingertips is a rich source for metabolic phenotyping**. A straight-forward workflow was established for sampling and processing sweat samples from fingertips. In short, hands are washed without soap and dried with a disposable paper towel prior to each sampling time-point. For sweat collection, a circular sampling unit standardised to 1.15 cm diameter was then held between thumb and index finger for 1 min and was transferred with clean tweezers into an empty tube for storage (Fig. 1a). The metabolites were extracted from the sampling units using aqueous conditions and the resulting solution was directly introduced into the liquid chromatography-mass spectrometry (LC-MS) system for analysis. Sample collection and processing required ~13 min per sample. Sampling can be performed by untrained personnel in a highly frequent manner and the non-invasive nature of the sampling facilitates patient compliance. Data acquisition requires a further 7.5 min, which gives a total of ~20 min for the entire workflow per sample.

Based on the known rates of sweat production in eccrine glands on the fingertips[29,40], the median sweat volume collected using this method can be estimated at around 200–2000 nL (2 min × 2 cm$^2$ × 50–500 nL min$^{-1}$ cm$^{-2}$) sweat per sample. High-resolution MS using a Q Exactive HF orbitrap hyphenated with an ultrahigh-performance liquid chromatography (UHPLC) system proved suitable for metabolic phenotyping from sweat samples (see methods). Initially, three participants were sampled multiple times in an observational study in order to evaluate the metabolic profile obtained from sweat of the fingertips of each individual. In detail, the participants collected sweat samples seven times per day at different intervals on 2 consecutive days and using both hands (see methods, study A). A total of 250 metabolites were identified and verified by external standards (Supplementary Data 1). Actually, many known as well as previously unknown endogenous and exogenous metabolites were identified in the sweat samples with high confidence (Fig. 1b, c). We detected not only a number of amino acid-related metabolites (e.g. tyrosine, leucine or citrulline), but also hormones (e.g. melatonin or progesterone). Newly identified metabolites include dopamine, progesterone and melatonin amongst others. Interestingly, we observed many coffee-derived metabolites, including caffeine and the related dimethyl– and methylxanthines. Principal component analysis (PCA) using those metabolites revealed that the samples clustered according to individuals (Fig. 1d). This indicated that the molecular composition of sweat associated with a given individual dominated the variances derived from multiple sampling. Interestingly, the principal components were strongly determined by the endogenous metabolites histamine, tryptophan, tyrosine and arginine (Supplementary Fig. 1). Moreover, we did not find notable

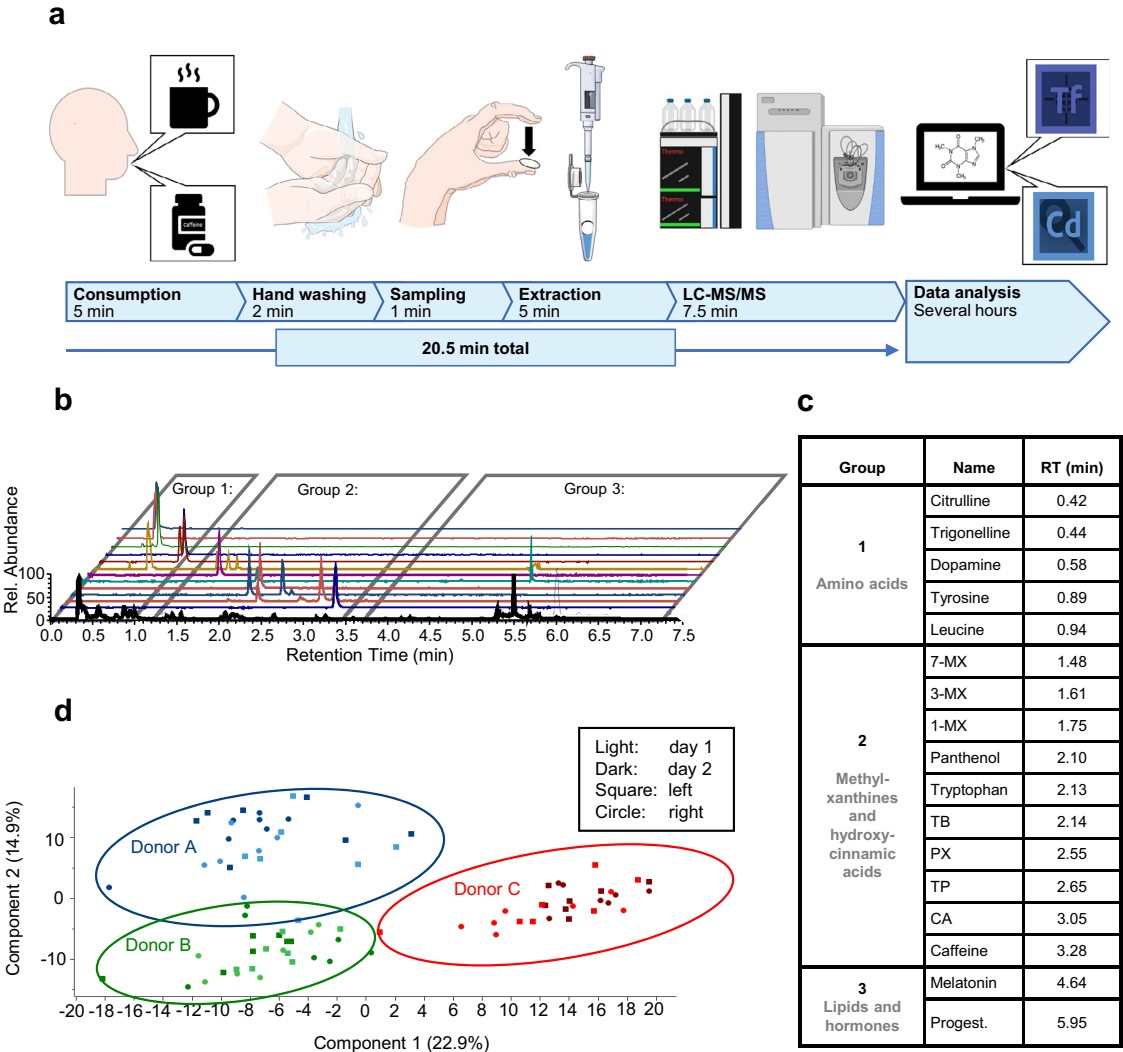

**Fig. 1 Sweat from the fingertips enables individualised metabolic biomonitoring.** A straight-forward workflow for sweat sampling and processing was established and successfully applied to proof-of-principle studies to investigate caffeine metabolism in an individualised fashion. **a** Graphical summary of the workflow including consumption of a cup of coffee or a caffeine capsule, sampling sweat from fingertips, the extraction of analytes and subsequent LC-MS/MS analysis as well as data analysis with respective durations in minutes. Panel **a** was modified from Servier Medical Art, licensed under a Creative Common Attribution 3.0 Generic License (http://smart.servier.com/) and BioRender (https://biorender.com/). Tf Tracefinder Software, Cd Compound Discoverer Software (both Thermo Fisher Scientific). **b** Extracted ion chromatograms of exemplary sweat components are shown. Based on their retention time, analytes were assigned to three groups as listed in **c**. **c** Identities of sweat constituents according to order of elution. CA chlorogenic acid, MX methylxanthine, PX paraxanthine, TB theobromine, TP theophylline, Progest. progesterone. **d** Principal component analysis (PCA) of finger sweat samples derived from the left (square) and right (circle) hand of three participants is depicted before and after coffee consumption at two different days (light and dark colour). PCA was calculated with a set of 250 metabolites (Supplementary Data 1) and successfully clustered the finger sweat samples according to the participants.

differences of the sweat composition between the left and right hand from a given individual (Fig. 1d).

**Sampling sweat from the fingertips is reliable and robust.** Biomolecules are characterised by LC-MS according to retention time (RT), the accurate mass of the molecular ion derived from the full mass spectrum (MS1) and the fragmentation pattern determined by tandem mass spectrometry (MS2). The experimentally determined mass-to-charge ratios of 15 representative metabolites showed mass deviations below <2 ppm, which are typical for Q Exactive HF instruments (Supplementary Table 1). The coefficient of variation (CV) of the RT determined for the internal standard caffeine-(trimethyl-D9) was found to be 1% across 636 injections (Fig. 2a, see methods, study A and C). Caffeine-(trimethyl-D9) was injected with every sample at 10 pg

on column. The CV of the areas under curve (AUCs) across the same sample set was 11% ($n = 636$). The CV improved slightly when considering study A only (CV = 7%, $n = 186$), but remained constant for study C (CV = 10%, $n = 450$). This indicated that the performance of the LC-MS system was robust across each sample set. MS2 spectra were of good quality and provided high matching factors, which supported the identification of previously known and newly identified metabolites found in sweat, e.g. tryptophan[42] and dopamine, respectively (Fig. 2b). Caffeine and its three main metabolites paraxanthine, theobromine and theophylline were spiked onto sampling units in the range of 1–100 pg µL$^{-1}$. These samples were processed according to the above-mentioned procedures and linear calibration curves were obtained with associated R$^2$ > 0.997 (Fig. 2c). At concentrations of 100 fg µL$^{-1}$, these molecules were still detected

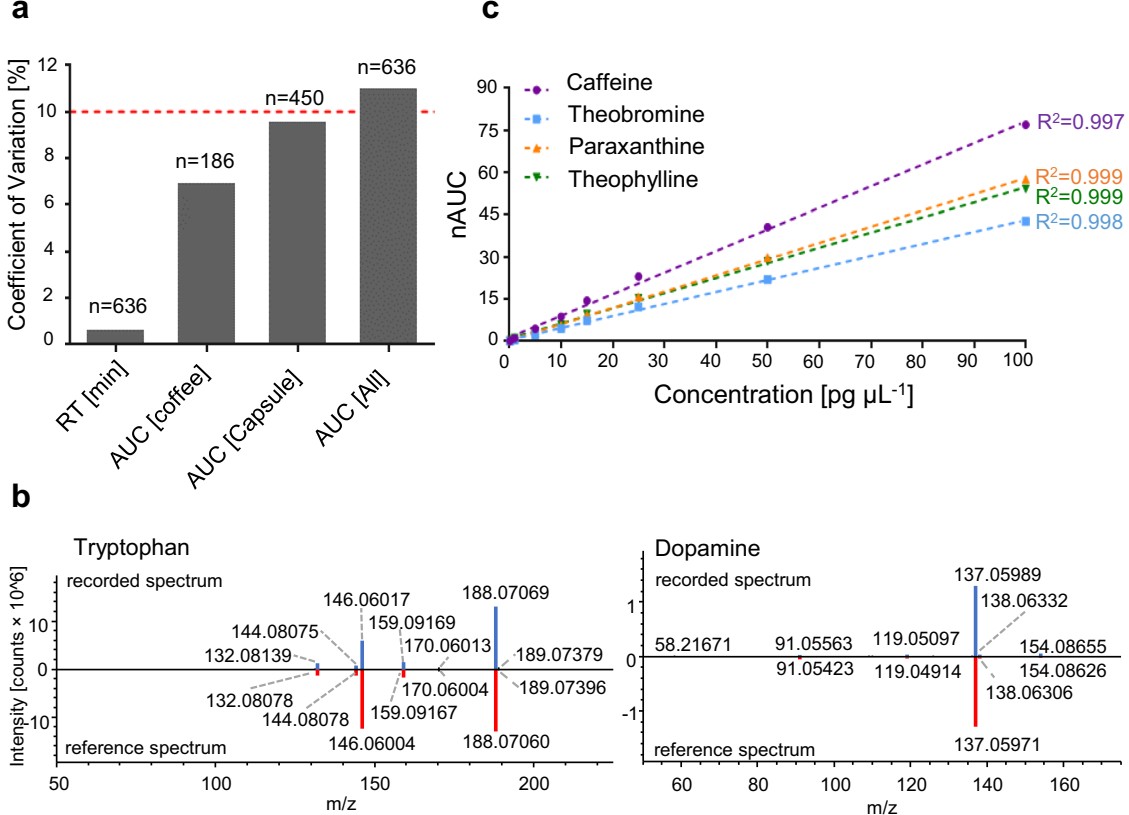

**Fig. 2 LC-MS/MS analysis of metabolites from sweat of the fingertips is precise and robust. a** Coefficients of variation of the retention times (RT) and areas under the curve (AUC) of a set of LC-MS/MS runs, as well as AUCs for the coffee (study B) and caffeine capsule (study C.1) intervention studies were determined for the internal standard caffeine-(trimethyl-D9). The means (boxes) and standard deviations are as follows: for the retention time $3.28 \pm 0.02$, for the coffee AUCs $1.80 \pm 0.13 \times 10^6$, for the capsule AUCs $1.56 \pm 0.15 \times 10^6$ and for all AUCs $1.63 \pm 0.18 \times 10^6$. The dashed red line was set to 10%. **b** Head-to-tail comparison of the recorded MS2 spectrum (blue) to the reference spectrum from *mzcloud* (red) of tryptophan (left) and dopamine (right) demonstrates high spectral quality supporting reliable compound identification. **c** Calibration curves for caffeine, theobromine, paraxanthine and theophylline with respective correlation factors ($R^2$) are shown. nAUC normalised area under the curve.

with signal-to-noise ratios >100 on the Q Exactive HF. Comparison of a spiked and processed caffeine standard ($10 \, \mathrm{pg \, \mu L^{-1}}$) to a directly injected caffeine standard ($10 \, \mathrm{pg \, \mu L^{-1}}$) yielded an extraction efficiency of 93%. The lower limit of quantification (LLOQ) was determined from the calibration curves as the mean AUC plus ten times the standard deviation of caffeine and its metabolites found in blank sampling units. This resulted in a LOQ of $0.2 \, \mathrm{pg \, \mu L^{-1}}$ for caffeine, $0.1 \, \mathrm{pg \, \mu L^{-1}}$ for paraxanthine and $1.7 \, \mathrm{pg \, \mu L^{-1}}$ for theobromine (see Source Data). The AUCs for theophylline in filter blanks and caffeine in tap water and paper towels were below the limit of detection (LOD), which was calculated as the mean AUC plus three times the standard deviation.

**Coffee consumption revealed coffee-specific xenobiotics in finger sweat.** After confirming sweat from the fingertips to contain endogenous metabolites, as well as xenobiotics mainly related to coffee consumption, we designed an intervention study with 11 participants, who consumed a standardised amount of coffee after a 12 h fasting period with regard to caffeine-containing food (see methods, study B). Two additional volunteers were sampled, who did not consume coffee, thus representing the control group. Sweat samples were collected before coffee consumption and subsequently after 15, 30, 45, 60, 90 and 120 min. Caffeine is a widely used stimulant of the central nervous system and features an excellent oral bioavailability[43,44]. Since the ingestion of an

equivalent of a double espresso was already shown to have systemic effects by affecting sleep behaviour[45–47], we expected to find caffeine and related xenobiotics upon coffee consumption in sweat from the fingertips. The metabolite levels of the participants before coffee consumption (0 min) revealed negligible amounts of chlorogenic acid, trigonelline and caffeine, while the primary metabolites of caffeine showed significant background levels (e.g. paraxanthine, theobromine and theophylline). The control group featured stable metabolite levels over time with small variations probably stemming from fluctuations in the rate of sweat excretion (Supplementary Fig. 2). Strikingly, the sweat from the fingertips 15 min post consumption revealed 35 xenobiotics of 121 metabolites (29%) contained in coffee presently identified by us from aqueous extracts of the roasted coffee beans used for this study, including among others caffeine, theobromine, theophylline, paraxanthine, methylxanthines, chlorogenic acid, trigonelline, methylsuccinic acid, quinic acid and iditol (Supplementary Data 2). The AUCs of caffeine, chlorogenic acid and trigonelline increased significantly in all volunteers as early as 15 min after coffee consumption (Fig. 3a). The time-dependent sampling revealed differences in kinetic properties of the coffee-specific xenobiotics, especially regarding absorption and clearance rates. For example, the AUCs of caffeine and chlorogenic acid peaked after 15 min, followed by rapid clearance, while the AUCs of the dimethylxanthines increased steadily over time on top of a pre-existing pool (Fig. 3b). Several coffee-specific metabolites

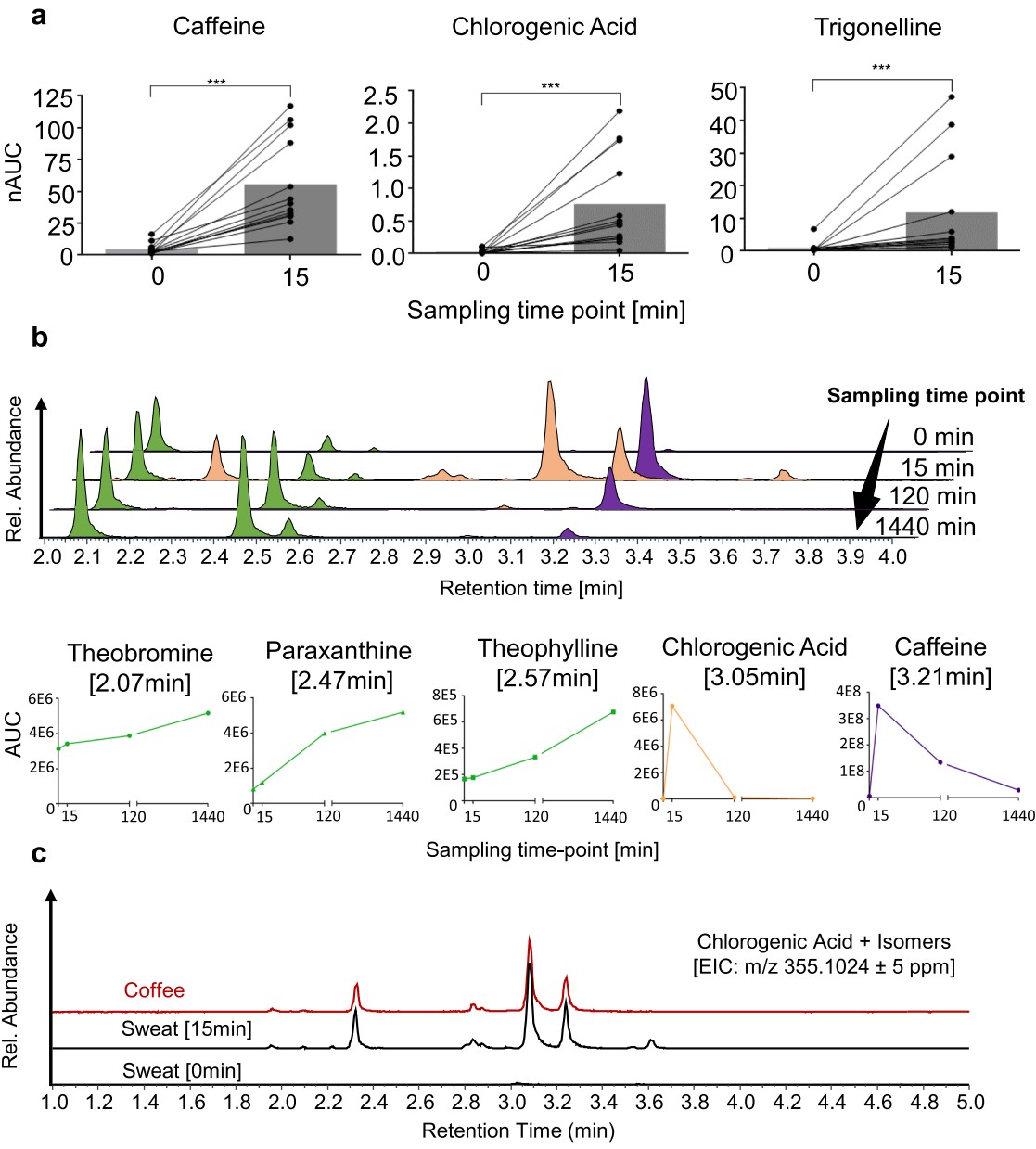

**Fig. 3 Xenobiotics are detected in a time-dependent manner in sweat from the fingertips after coffee consumption. a** Levels of normalised areas under the curve (nAUCs) for caffeine, chlorogenic acid and trigonelline, before (0) and 15 min (15) after coffee consumption are shown, demonstrating a significant increase in all participants ($n = 13 \times 2$ time-points) after 15 min. D'Agostino & Pearson test was performed to check normality of the data. Paired two-tailed Wilcoxon signed rank tests were performed for 13 volunteer profiles, delivering a $p$-value = 0.0002 for caffeine, chlorogenic acid and trigonelline. The mean nAUCs (boxes) and standard deviations are the following: for caffeine $4.8 \pm 4.4$ at 0 min and $56 \pm 35$ at 15 min, for chlorogenic acid $0.03 \pm 0.04$ at 0 min and $0.8 \pm 0.7$ at 15 min, for trigonelline $1.0 \pm 1.7$ at 0 min and $12 \pm 16$ at 15 min. **b** The temporal evolution of metabolite profiles is exemplarily shown for one participant (Volunteer 3, study A). The sampling time-point 1440 min represents the time-point before consumption on the second sampling day. Whereas caffeine (violet) and chlorogenic acid AUCs (orange) were found to increase quickly after coffee consumption followed by rapid clearance, the levels of theobromine, paraxanthine and theophylline (green) increased more slowly within the observation period. **c** Similarity of extracted ion chromatograms (EIC) of chlorogenic acid and its isomers from coffee extracts and from sweat of the fingertips 15 min after coffee consumption. The corresponding sample collected just before coffee consumption (0 min) served as negative control.

displayed a number of isomers in their extracted ion chromatograms. For example, chlorogenic acid ($m/z$ 355.1024, RT = 3.05 min) showed at least five isomers (Fig. 3c) as verified on MS2 level. The ratio of the relative peak intensities of chlorogenic acid and its isomers was conserved when comparing coffee extracts and sweat from the fingertip. This indicated that these isomers are equally distributed into the water-soluble body compartment and are equally cleared from body on a rapid timescale. Chlorogenic

acids and its isomers were not observed prior to coffee consumption. Such a comparative analysis strategy may be used to discover other xenobiotics distributed to sweat glands in a systemic fashion as indicated by the yet unidentified feature detected at $m/z$ 337.0920 (Supplementary Fig. 3). These findings provide evidence that ingested xenobiotics may be robustly detected in the sweat from the fingertips, and their time-dependence mirrors their pharmacokinetic properties.

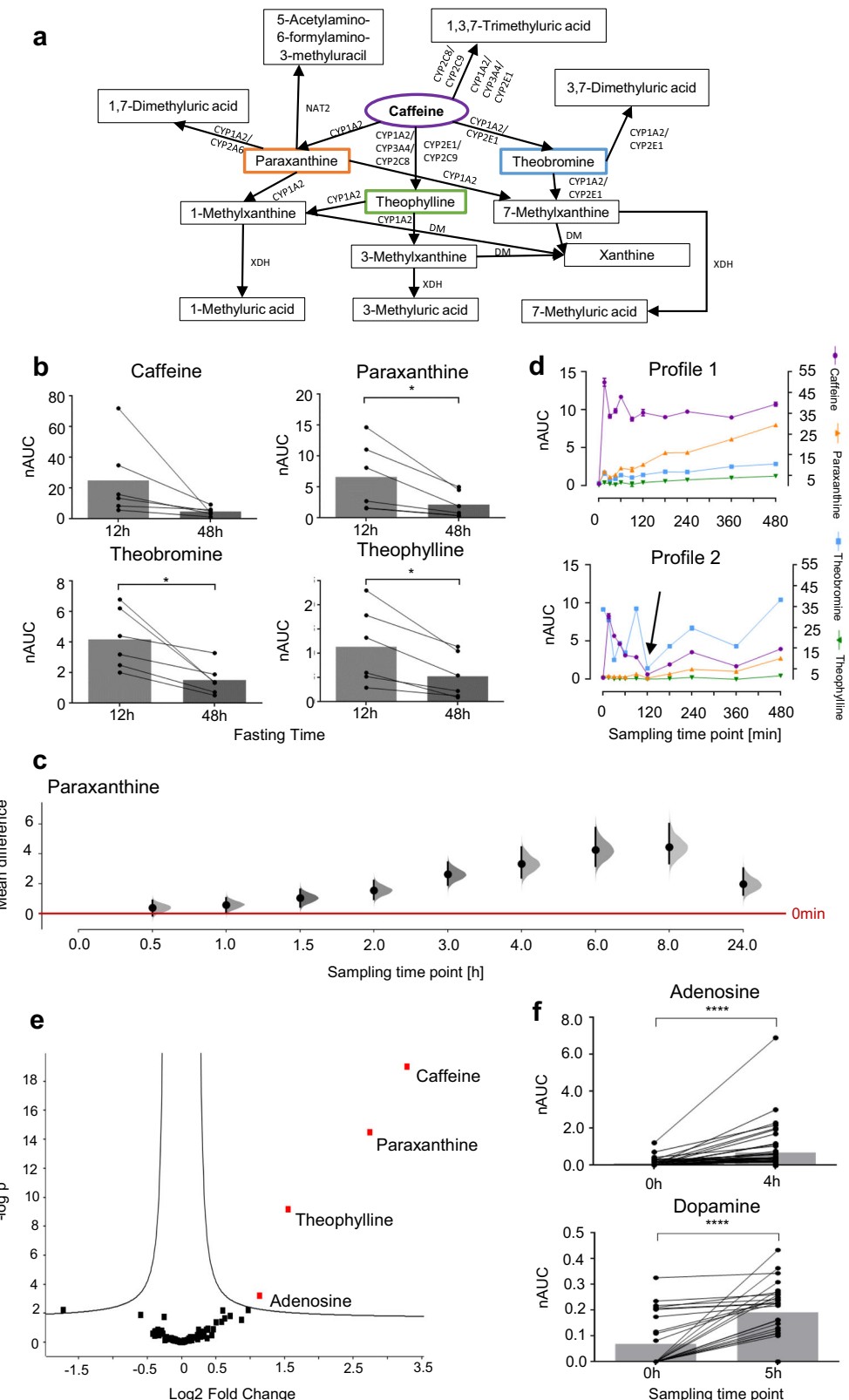

**Finger sweat enables to elucidate individual metabolic traits**. The metabolism of caffeine by different hepatic enzymes is well known[48], and the catabolic products were successfully identified in sweat from fingertips after coffee consumption (Fig. 4a, Supplementary Table 1). However, dimethyl– and methylxanthines may originate from both coffee beans and from endogenous hepatic metabolism. Additionally, we observed significant background levels of these metabolites in sweat from the fingertips before coffee consumption. In order to monitor the physiological conversion of caffeine into dimethylxanthines by hepatic enzyme

**Fig. 4 Consumption of a caffeine capsule enables to elucidate individualised metabolic traits from sweat of the fingertips. a** Caffeine metabolism including known metabolic routes, metabolites and related enzymes: CYP cytochromes P450, NAT2 N-acetyltransferase 2, XDH xanthine dehydrogenase, DM demethylase. These metabolites were all detected in sweat from the fingertips. **b** Six individuals participated in the coffee as well as in the caffeine capsule studies. The AUCs of caffeine and the primary metabolites are compared depending on the duration of the fasting period (12 vs 48 h, $n = 6$). Longer fasting significantly reduced the amounts of xenobiotics in sweat from the fingertips. It was tested with Kolmogorov–Smirnov test using Dallal–Wilkinson-Lilliefors $p$-value if values came from a Gaussian distribution. A two-tailed paired $t$-test (6 participants × 2 time-points) was performed for caffeine, paraxanthine, theobromine, and theophylline. **c** Shared-control plot with data from 47 volunteer profiles for paraxanthine is shown. The mean differences between the control group (time-point before consumption, red line) and each of the sampling time-points post ingestion is plotted on the $y$-axis. Paraxanthine is significantly upregulated from the sampling time-point 1.5 h on after ingesting a caffeine capsule. The effect size is presented as a bootstrap 95% confidence interval. Mean difference, lower and upper limits are provided in the Source data. **d** Exemplary metabolic profiles of two participants, demonstrating individual differences in metabolic properties regarding caffeine metabolism as exemplified by the preferential formation of paraxanthine in volunteer profile 1 in contrast to theobromine in case of volunteer profile 2. Caffeine is displayed on the right $y$-axis, while theobromine, paraxanthine and theophylline are displayed on the left $y$-axis. Error bars represent standard deviation of two technical replicates ($n = 2$) for each of the 11 time-points. Means and standard deviations can be found in the Source data. **e** Metabolic changes 4 h after consuming a caffeine capsule demonstrated with a volcano plot illustrating the similarities of metabolite regulations in 47 volunteer profiles. Next to the known caffeine metabolites, adenosine is regulated. **f** Boxplots for adenosine and dopamine before and 4 h/5 h after consuming a 200 mg caffeine capsule shown for 47 (study C.1 and C.2)/ 27 (study C.2) volunteer profiles. Normality of the data was checked with D'Agostino-Pearson test. A two-tailed Wilcoxon Signed Rank Test was performed for adenosine. A tow-tailed $t$-test was performed for dopamine. nAUC normalised area under the curve. Boxes represent the means of each time-point. All statistical test results as well as means and standard deviations can be found in the methods section.

activity, we designed a study in which participants refrained from consuming caffeine-containing products for at least 48 h before ingesting a single caffeine capsule (200 mg). The caffeine capsule and the longer fasting time were chosen to minimise background contributions from catabolic products of caffeine. Forty volunteers were enrolled in this study and sweat from the fingertips was sampled repeatedly over 27 h with up to 20 sample collections per volunteer (see methods, study C.1 and C.2). Six individuals participated in both the coffee consumption study (study B) and the caffeine capsule study (study C.1). Indeed, their prolonged fasting featured an improved baseline and revealed a significant decrease of dimethylxanthines to negligible levels after the 48 h fasting period compared to the 12 h fasting period (Fig. 4b). Ingestion of the caffeine capsule significantly increased the abundance of caffeine in sweat from fingertips in all volunteers already after 15 min, in accordance with coffee consumption. The caffeine abundance remained elevated for at least 480 min in all volunteers and returned close to baseline after 24 h (Supplementary Fig. 4). The abundance of the primary metabolite paraxanthine increased more slowly and peaked between 360 and 480 min post ingestion (Fig. 4c). Individual metabolic time-courses revealed rather striking differences regarding caffeine metabolism (Fig. 4d). For example, volunteer profile 1 displayed a sharp increase in caffeine abundance, which remained relatively constant over 480 min, while paraxanthine abundance increased steadily during this time period. In contrast, volunteer profile 2 featured a similar increase in caffeine abundance, but started with an elevated theobromine baseline, which also represented the main metabolite of caffeine. These findings suggest that sampling sweat from the fingertips may be of particular interest for characterising personalised metabolic traits. Cytochrome P450 enzymes are key players in the hepatic metabolism and several isoforms are known to process xenobiotics at different rates[48]. Thus, xenobiotics like caffeine may be subjected to variable metabolisms depending on the individual expression of these enzymes. This may reveal individual physiological responses to xenobiotic exposure that may serve as proxies for hepatic metabolic activity. Therefore, the influence of the metabolic turnover of caffeine depending on the expression of cytochrome P450 enzymes was investigated in vitro using HepG2 cells (Supplementary Information, Supplementary note 1). Indeed, we found that HepG2 cells would increase the metabolic turnover of caffeine to its primary metabolites upon chemical induction of cytochrome P450 enzymes with benzo-[a]-pyrene (Supplementary Fig. 6). Moreover, the induction of these

enzymes also affected the relative ratios of the primary metabolites significantly. This supports the conclusion that the individual enzymatic activity status may modulate the formation of metabolites subsequently detected in sweat from the fingertips. Statistical analysis of the metabolites reproducibly detected in all 47 (study C.1 + C.2) or 27 (study C.2) volunteer profiles revealed the significant upregulation of caffeine, paraxanthine and theophylline, as well as adenosine 4 h post ingestion. Theophylline and paraxanthine reflected the metabolic turnover of caffeine within each volunteer profile, while adenosine was identified as an endogenous metabolite upregulated upon caffeine ingestion (Fig. 4e, f). Another endogenous metabolite, dopamine was significantly induced 5 h after consuming a caffeine capsule in 27 participants (study C.2, Fig. 4f, Supplementary Fig. 5). Adenosine and dopamine are not directly related to caffeine metabolism.

**Mathematical modelling quantifies individual dynamic metabolic responses.** Fluctuations in the rate of sweat excretion cause significant variance in the collected sweat volumes. This represents a fundamental challenge for the time-course analysis of sweat from the fingertips. For example, the apparent down-regulation of all analytes at 120 min in volunteer profile 2 (Fig. 4d, arrow) strongly suggests that at that time-point less sweat was collected in comparison to the adjacent measurements (see Fig. 5e, arrow). Moreover, the magnitude of this effect on the apparent concentration is unknown. We used dynamic metabolic network modelling to discern the effects of the sweat volume on the measured time-series of caffeine catabolism in the body (see methods). In brief, caffeine uptake and clearance via its major metabolic products paraxanthine, theobromine and theophylline can be described by first order kinetics (Fig. 5a)[49,50]. Due to fasting we can set the initial caffeine concentration at time 0 min to zero (Fig. 4b). Additionally, we consider the sweat volume to be a function of time, but assume that at every time-point the sweat volume is constant across all metabolites. The assumption holds if the modelled metabolites are not reabsorbed during sweating. The resulting mathematical model was fitted to each volunteer. We estimated the kinetic constants, the initial concentrations of paraxanthine, theobromine and theophylline and the sweat volumes at each time-point, as exemplified for volunteer profiles 1 and 2 (Fig. 5b, c, e, f and Supplementary Table 2). In both cases our model accurately described individual caffeine metabolisms with good accuracy (goodness of fit $R^2_{adjusted} > 0.90$). Besides the possibility to estimate the rate of sweat excretion by means of

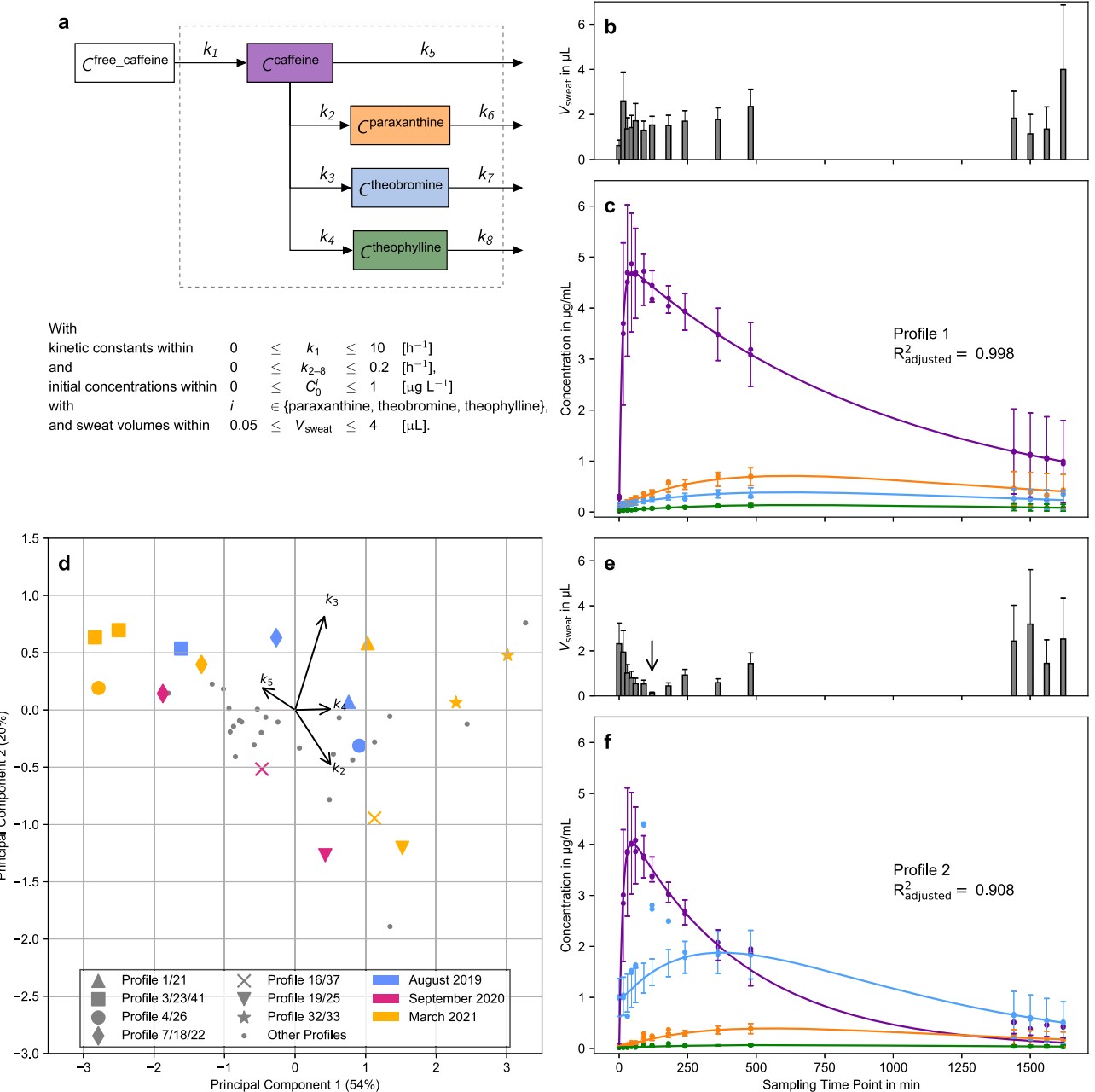

**Fig. 5 Metabolic networks facilitate the discovery of dynamic metabolic patterns from individuals. a** Network of caffeine and its major metabolites that was used for fitting of the concentration time-series and the constraints of fitting parameters. **b**, **e** Estimated sweat volumes ($V_{sweat}$) in µL of the measurements of volunteer profiles 1 and 2, respectively. Each bar represents a single $V_{sweat}$. The error bars show the 95% confidence intervals of the model prediction. **c**, **f** Fitted concentration time-series of caffeine, paraxanthine, theobromine and theophylline for volunteer profiles 1 and 2 (compare Fig. 4d). The lines refer to the fitted concentration and the symbols refer to the measured values ($\tilde{\mathbf{M}}$) divided by $V_{sweat}$ at each time-point (Eq. (1)). The error bars show the 95% confidence intervals of the model prediction ($n = 2$ technical replicates times 4 metabolites times 15 time-points per profile). The arrow marks the sweat volume of profile 2 at 120 min (see text). A visual representation of the influence of the sweat volume on the fit is shown in Supplementary Fig. 7. **d** Two-dimensional PCA plot of the fitted caffeine conversion constants. Volunteer profiles (i.e. time-series) from the same participant are plotted with large, coloured symbols whereas participants who contributed only once are marked with small grey circles. The colours represent the month of sampling.

this modelling approach, the shape of the curves visualises the dynamic metabolic patterns of each individual.

Interestingly, the kinetic constants for uptake ($k_1$) of caffeine is within the standard deviation, while the constants of conversion ($k_2$, $k_3$, $k_4$) are approximately half of the literature values of population averages for blood plasma (Supplementary Table 2)[51]. Whereas the fractional conversion of caffeine to the main metabolic product paraxanthine in volunteer profile 1 is similar

to what is described as population average[51,52] we saw substantial differences for volunteer profile 2, who displayed theobromine as the main metabolic product of caffeine (Supplementary Table 3). We found individual differences to be robust over time. In Fig. 5d a two-dimensional PCA plot of the fitted conversion constants of caffeine ($k_2$, $k_3$, $k_4$, $k_5$) is shown. Individuals who generated at least two volunteer profiles (i.e. independent time-series) are marked with large symbols. Their respective colour indicates the

month of sampling. Not only do two profiles of the same volunteer within one month cluster close to each other (e.g. star symbols), but also the ones that were sampled more than 1.5 years apart are in close proximity (e.g. diamond symbols). The biggest difference of volunteer profiles from one participant was found for profiles 4 and 26 (big circles). For volunteer profile 26, however, we observed an overall poor fit ($R^2_{adjusted}$ of 0.56 compared to 0.984 for profile 4). On another note, the original axes in Fig. 5d show that the catabolism of caffeine into paraxanthine ($k_2$) and direct elimination ($k_5$) is negatively correlated, whereas the catabolism of caffeine into theobromine ($k_3$) and theophylline ($k_4$) is positively correlated. This correlation is known in literature and is likely due to common hepatic cytochrome P450 enzymes catalysing the conversion of caffeine to theobromine and theophylline (Fig. 4a)[53].

**Targeted assays can be established for clinical implementation**. The described metabolic phenotyping approach represents a powerful discovery tool for endogenous and xenobiotic compounds found in sweat of the fingertips. In order to evaluate the feasibility of clinical implementation, we established a targeted assay for caffeine, and the primary metabolites theobromine, theophylline and paraxanthine on a triple quadrupole MS using multiple reaction monitoring (MRM, see Supplementary Information, Supplementary note 2 and Supplementary Table 4). For this purpose, five participants consumed a standardised coffee on 3 independent days after a 12 h caffeine-free fasting period and samples were collected at different time intervals in analogy to study B. The assay was validated and revealed linear ranges between 0.5 and 300 pg µL$^{-1}$ of the respective metabolites (0.25–150 pg on column, Supplementary Fig. 9). LOD values were <0.2 pg µL$^{-1}$ per collected sweat sample. The overall process efficiencies were generally >88% and the precision of 25 pg µL$^{-1}$ spiked metabolite was <1% (Supplementary Table 5), while the overall CV of the AUC of caffeine 5 h after coffee consumption of all volunteers over 3 independent days was 22% (Supplementary Fig. 10). This suggests that targeted assays based on the analysis of sweat from the fingertips can be successfully established directly from metabolic phenotyping.

## Discussion

The present study provides evidence that sweat from the fingertips can be used for dynamic metabolic phenotyping. The sample collection is non-invasive, safe and can be accomplished by untrained personnel, supporting patience compliance[47]. Other minimally to non-invasive approaches such as microneedle patches or sweat patches, require longer collection periods of several minutes up to days, aiming to collect sweat at a single timepoint[17,54,55]. In our approach, time-course analyses with frequent sampling can be performed due to the facile collection procedure. Our setup allows the analysis of unstimulated sweat in contrast to published approaches where sweat production was induced with pilocarpine iontophoresis (coupled with the Macroduct sweat collector) or physical exercise. Such stimuli were shown to alter the physiological sweat composition, which may introduce bias into the analysis[17,56]. The entire workflow can be accomplished within 20 min per sample, and has the potential to support large scale longitudinal metabolic studies. However, metabotyping the small amounts of sweat requires sufficiently sensitive analytical equipment. Although our approach centres on metabolic profiling using dedicated high-resolution instrumentation, we demonstrated the successful transfer to a targeted assay. Targeted MS is now routinely implemented in the clinical laboratory[57].

Sweat from the fingertips represents a rich source for metabolic phenotyping. Considering that a given metabolite may be represented in an LC-MS experiment by several features due to different adducts and charge states[58], it may be estimated that several thousand distinct metabolites can potentially be identified in sweat from the fingertips using this methodology. So far, we have verified 250 metabolites with external standards (Supplementary Data 1). The analysis is robust and sensitive with limits of detection of metabolites found in the sub-picogram range per sweat sample. Indeed, the detection limits found in this study showed improved sensitivity compared to previously used methodologies[59]. As a result, numerous endogenous metabolites were identified, which have not yet been described in sweat, including dopamine, progesterone and melatonin (Fig. 1). This highlights the potential of this approach to successfully identify low-abundant metabolites, which are challenging to detect in other biofluids due to matrix effects (e.g. melatonin in blood or plasma)[59–61]. Analysis of the area under the curve of the internal standard revealed an overall coefficient of variation of 11% across 636 samples and indicated acceptable precision (Fig. 2).

Proof-of-principle intervention studies were successfully carried out and support the applicability of the method. In two separate studies, participants were asked to consume a standardised cup of coffee or ingest a caffeine capsule after a caffeine– and theobromine-free diet for 12–72 h. After ingestion, sweat samples were collected up to 20 times within 27 h per volunteer. Sampling intervals of 15 min were feasible. Coffee consumption led to a significant upregulation of caffeine, chlorogenic acid and trigonelline within 15 min in all participants (Fig. 3, study B). This suggested a fast absorption and distribution of these xenobiotics, which also displayed distinct absorption and excretion kinetics (Fig. 3c). Altogether, 35 metabolites originating from coffee were detected in sweat from the fingertips.

The observation of significant background levels of dimethylxanthines after coffee consumption in study B pointed towards a confounding problem with respect to the origin of these caffeine metabolites. In fact, their temporal increase may have been due to their absorption from consumed coffee and hepatic caffeine metabolism. In order to resolve this question, we designed an additional study in which participants ingested a caffeine capsule (200 mg) only and adhered to a longer caffeine– and theobromine-free fasting regime. Of note, the longer fasting periods (48–72 h) significantly reduced the background levels of the primary metabolites (Fig. 4b, study C) compared to 12 h fasting (study B). Interestingly, statistical analysis of the metabolic profiling data from study C, involving the caffeine capsule, revealed a significant upregulation of caffeine and of the metabolic products theophylline and paraxanthine across all participants after 480 min (Fig. 4). Moreover, participants featured significantly increased levels of dopamine after 5 h. Being an endogenous metabolite, it is plausible to assume that this upregulation corresponded to a physiological response to caffeine ingestion. Increased dopamine levels were already observed upon caffeine[62], as well as coffee consumption by others[63]. Adenosine was significantly induced 4 h post ingestion of a caffeine capsule. Caffeine exerts most of its biological actions such as countering sleep pressure via antagonising adenosine receptors[64]. It has been demonstrated that caffeine increases plasma adenosine concentration potentially via receptor-mediated regulation of the plasma adenosine concentration[65] and this finding seems to extend to sweat from the fingertips. We have previously described individual opposing responses with regard to anti-inflammatory effects after coffee consumption[66]. Such studies required the collection of blood from volunteers and this could now be facilitated by analysing sweat from the fingertips. Adenosine is also known to be an anti-inflammatory mediator that may regulate neutrophils, macrophages and lymphocytes through interacting with surface receptors of these cells[67]. It is important

**Table 1 Overview of the three studies discussed in this publication.**

| Study | Volunteers | Design | Fasting [hours] | Sampling time-points |
|---|---|---|---|---|
| A | 2 males, 1 female | Observational | 12 | 0, 15, 30, 45, 60, 90 and 120 min on 2 consecutive days |
| B | 7 males, 6 females | Double espresso or control group | 12 | 0, 15, 30, 45, 60, 90 and 120 min |
| C.1 | 8 males, 9 females | Caffeine capsule | 48 | 0, 15, 30, 45, 60, 90, 120 min and 3, 4, 6, 8, 24, 25, 26 and 27 h |
| C.2 | 16 males, 11 females | Caffeine capsule | 72 | 0, 15, 30, 45, 60, 90, 120 min and 3, 4, 5, 6, 7, 8, 9, 10, 11, 12, 13, 14 and 24 h |

to note that sweat from the fingertips may not only reveal ingested xenobiotics, but also endogenously produced metabolic products and physiological responses to bioactive xenobiotics.

Individual metabolic traits were then investigated by analysing the time-dependent metabolic evolution of caffeine upon ingesting a caffeine capsule (Fig. 4d). We found that sweat from the fingertips may be successfully used for the personalised assessment of such metabolic activities. Importantly, this strategy may be extended to other xenobiotics or drugs and their causally related metabolic products in order to obtain insight into specific processes of human metabolism in an individualised manner. Moreover, by inducing cytochrome P450 enzymes in HepG2 cells in vitro (Supplementary note 1), we were able to modulate the metabolic turnover of caffeine and the formation of specific catabolic products. This suggests that the relative ratios of caffeine to its primary metabolites may reflect hepatic activity, since the physiological hepatic metabolism of caffeine relies on a similar set of enzymes as in HepG2 cells.

Variations in the sweat volume over the course of the study represented a major challenge for normalisation and quantification. Mathematical modelling overcame this issue by addressing molecular constraints of substrate-product relations of enzymatically linked metabolites. Successful modelling has two central prerequisites: firstly, the measurement of at least two metabolites with known dynamics and, secondly, a linear relationship of said metabolites to the sweat rate. Importantly, this allowed us to compute a sweat volume that is proportional to all metabolites at each time-point. This approach was capable of delivering estimates of individual rate constants for drug uptake, metabolism and clearance and therefore allows to model dynamic metabolic patterns in individuals (Fig. 5). Sampling sweat from the fingertips enables time-course studies, which are evaluated by means of conversion rates of metabolically related substance classes. Their observed robustness suggests that the development of personalised tests via finger sweat measurements is feasible. For example, caffeine elimination was shown to be a proxy for liver function[68], and we hypothesise that a future study using an experimental setup identical to the caffeine capsule study could differentiate between patients with cirrhotic and normal livers. Additionally, we argue that the method presented here provides a convenient solution to the normalisation problem of finger sweat, which previously only has been tackled by employing microcapillaries[69]. However, they require large volumes of sweat, and thus need either long sampling times or require physical exercise. Both are detrimental when measuring fast pharmacokinetics, for example, for caffeine this would circumvent the requirement of absolute quantitative information of a single measurement.

In summary, metabolic phenotyping from sweat of the fingertips in conjunction with mathematical modelling is a promising approach to obtain dynamic metabolic patterns from individuals that may overcome the limitations of conventional composition biomarkers. Further research is currently performed in order to consolidate the potential of sampling sweat from the fingertips for applications in precision medicine.

## Methods

**Reagents and chemicals.** LC-MS grade methanol, water, acetonitrile and formic acid used during sample preparation and LC-MS/MS analysis were purchased from VWR chemicals (Vienna, AT). Xenobiotic and metabolite standards (caffeine, theobromine, theophylline, paraxanthine, 1-methylxanthine, 3-methylxanthine, 7-methylxanthine, 1-methyluric acid, 3-methyluric acid, 1,7-dimethyluric acid, 3,7-dimethyluric acid and 1,3,7-trimethyluric acid, chlorogenic acid, xanthine, 5-Acetylamino-6-formylamino-3-methyluracil, dopamine and proteinogenic amino acids) were either purchased from Sigma–Aldrich (Vienna, AT) or Honeywell Fluka (GER). Caffeine capsules were bought from Mach dich wach! GmbH (GER). Sampling units were made from filter papers (precision wipes, number = 7552, white, 11 × 21 cm, Kimtech Science, Kimberly-Clark Professional, USA) using a circular puncher of 1cm².

**Standard solutions and calibration samples.** Stock solutions of 1 mg mL⁻¹ of the analytical standards and the internal deuterated standards caffeine-(trimethyl-D9) and N-acetyl-tryptophan in methanol were prepared and stored at 4 °C. For caffeine, paraxanthine, theobromine and theophylline calibration curves were generated by spiking onto sampling units with the following concentrations: 0.1, 1, 5, 10, 15, 25, 50 and 100 pg µL⁻¹. The internal deuterated standards were prepared at a concentration of 1 pg µL⁻¹ in an aqueous solution containing 0.2% formic acid, which served as the extraction solution for all samples.

**Cohort design.** Altogether, 21 males and 19 females with ages between 20–55 years and a BMI of 21 ± 8 kg m⁻² were enrolled in this study. Participants had different dietary habits regarding the consumption of coffee; rare to regular consumption. Prior sampling, participants were required to fast caffeinated food (e.g. chocolate) and drinks (e.g. coffee, tea and energy drinks) for a period of 12–72 h. Sweat samples from the fingertips were collected at different time intervals and in the presence or absence of an intervention (see Table 1, studies A–C). Study B involved the consumption of a standardised coffee (equivalent to a double espresso), while studies C.1 and C.2 involves the ingestion of a caffeine capsule (200 mg). Seven volunteers have participated in more than one study, which gave a total of 47 volunteer profiles for study C. It was ensured that the volunteers did not touch the prepared coffee with their fingers.

**Collection of sweat from the fingertips.** Sampling units of 1 cm² circular surface were pre-wetted with 3 µL water and provided in 0.5 mL Eppendorf tubes. For each sweat collection, volunteers cleaned their hands using warm tap water and dried them with disposable paper towels. Volunteers kept their hands open in the air at room temperature for 1 min. Then, the sampling unit was placed between thumb and index finger using a clean tweezer and held for 1 min. Sweat formation was not forced. Filters were transferred back to labelled 0.5 mL Eppendorf tubes using a clean tweezer and stored at 4 °C until sample preparation.

**Sample preparation.** Coffee extracts were prepared taking an aliquot of 1 mL of a 250 mL coffee cup used for study A and B, which was centrifuged for 10 min at 15000 × g. The supernatant was diluted 1:100, 1:1000 and 1:10000 with the extraction solution consisting of an aqueous solution of caffeine-(trimethyl-D9) (1 pg µL⁻¹) with 0.2% formic acid. The dilutions were again centrifuged before analysis by LC-MS/MS.

For the extraction of metabolites from the sampling units, 120 µL of the extraction solution consisting of an aqueous solution of caffeine-(trimethyl-D9) (1 pg µL⁻¹) with 0.2% formic acid was added into the 0.5 mL Eppendorf tube containing the sampling unit. The metabolites were extracted by pipetting up and down 15 times. The sampling unit was pelleted on the bottom of the tube and the supernatant was transferred into HPLC vials equipped with a 200 µL V-shape glass insert (both Macherey-Nagel GmbH & Co.KG) and analysed by LC-MS/MS. Additionally, 10 unused filter, 10 paper towels and 10 tap water blanks were extracted similarly to determine potential contaminants and metabolite background levels.

**LC-MS/MS analysis.** A Q Exactive HF (Thermo Fisher Scientific) mass spectrometer coupled to a Vanquish UHPLC System (Thermo Fisher Scientific) was employed for this study. Chromatography was performed using a Kinetex XB-C18

column (100 Å, 2.6 μm, 100 × 2.1 mm, Phenomenex Inc.). Mobile phase A consisted of water with 0.2% formic acid, mobile phase B of methanol with 0.2% formic acid and the following gradient program was run: 1–5% B in 0.3 min and then 5–40% B from 0.3–4.5 min, followed by a column washing phase of 1.4 min at 80% B and a re-equilibration phase of 1.6 min at 1% B resulting in a total runtime of 7.5 min. Flow rate was set to 500 μL min$^{-1}$, the column temperature to 40 °C, the injection volume was 10 μL and the injection peak was found at RT = 0.3 min. All samples were analysed in technical duplicates. An untargeted mass spectrometric approach was applied for compound identification. Electrospray ionisation was performed in positive and negative ionisation mode. MS scan range was $m/z$ 100–1000 and the resolution was set to 60000 (at $m/z$ 200). The four most abundant ions of the full scan were selected for HCD fragmentation applying 30 eV collision energy. Fragments were analysed at a resolution of 15000 (at $m/z$ 200). Dynamic exclusion was applied for 6 s. The instrument was controlled using Xcalibur software (Thermo Fisher Scientific).

**Data analysis**. Raw files generated by the Q Exactive HF instrument were analysed using the Compound Discoverer Software 3.1 (Thermo Fisher Scientific). Identified compounds were manually reviewed using Xcalibur 4.0 Qual browser and Freestyle (version 1.3.115.19) (both Thermo Fisher Scientific) and the obtained MS2 spectra were compared to reference spectra, which were retrieved from *mzcloud* (Copyright © 2013–2020 HighChem LLC, Slovakia). The match factor cut-off from *mzcould* was 80, while the mass tolerances were 5 and 10 ppm on MS1 and MS2 levels, respectively. Moreover, the identity of compounds suggested by Compound Discoverer was verified by analysing purchased standards using the same LC-MS method. The Tracefinder Software 4.1 (Thermo Fisher Scientific) was used for peak integration and calculation of peak areas. The generated batch table was exported and further processed with Microsoft Excel (version 1808), GraphPad Prism (version 6.07) and the Perseus software (version 1.6.12.0)[70], the letter being used for the principal component analysis. Untargeted metabolic profiling by mass spectrometry delivered more than 50000 reproducible sweat-specific features per analysis. Microsoft PowerPoint (version 1808) was used for creating figures.

**Statistical analysis**. D'Agostino-Pearson tests as well as Kolmogorov–Smirnov tests with Dallal–Wilkinson–Lilliefors p-value were performed to test if values came from a gaussian distribution. Two-tailed, paired t-tests or Wilcoxon Signed Rank Tests were performed for mass spectrometry data using GraphPad Prism (Version 6.07) to evaluate the significance of the abundance increase/decrease of compounds and their metabolites. For Fig. 4b it was tested with Kolmogorov–Smirnov test using Dallal–Wilkinson–Lilliefors p-value if values came from a Gaussian distribution. A two-tailed paired t-test (6 participants × 2 time-points) for caffeine (p-value = 0.1033, t = 51.990, df = 5), paraxanthine (p-value = 0.0297, t = 3.012, df = 5), theobromine (p-value = 0.0203, t = 3.353, df = 5) and theophylline (p-value = 0.0118, t = 3.866, df = 5). Means and standard deviations are for caffeine 25 ± 25 for 12 h fasting and 4.8 ± 2.7 for 48 h fasting, for paraxanthine 6.6 ± 5.5 for 12 h fasting and 2.2 ± 2.1 for 48 h fasting, for theobromine 4.2 ± 2.0 for 12 h fasting and 1.5 ± 1.0 for 48 h fasting, for theophylline 1.1 ± 0.8 for 12 h fasting and 0.5 ± 0.5 for 48 h fasting. For Fig. 4f normality of the data was checked with D'Agostino-Pearson test. A two-tailed Wilcoxon Signed Rank Test was performed for adenosine (n = 47, sum of positive ranks = 1020, sum of negative ranks = −17,00, sum of signed ranks = 1003, p-value ≤ 0.0001). A tow-tailed t-test was performed for dopamine (p-value ≤ 0.0001, t = 5.416, df = 26). The means and standard deviations are the following: for adenosine 0.1 ± 0.2 at 0 h and 0.7 ± 1.2 at 4 h, for dopamine 0.1 ± 0.1 at 0 h and 0.2 ± 0.1 at 5 h. Volcano plots were obtained using Perseus Software[70], setting the false discovery rate (FDR) to 0.05 and the minimal fold change (s0) to 0.1. For Fig. 4e the −log p-value for caffeine is 19.02, for paraxanthine 14.48, for theophylline 9.16 and for adenosine 1.14. Shared-control plots were generated with an R script[71].

**Mathematical modelling**. The model describes the concentration time-series of the ingested free caffeine and four sweat metabolites (caffeine, paraxanthine, theobromine, theophylline) within the constraints of following assumptions (Fig. 5a):

- caffeine metabolism can be described by mass-action kinetics in a one-compartment body model[49,50],
- the uptake of external caffeine is instantaneous (i.e. no lag time between ingestion and absorption into the body),
- the steady-state volume of distribution of caffeine, paraxanthine, theobromine and theophylline is instantaneously reached and time independent[50,51],
- concentration enrichment due to an increase in the water fraction from blood to sweat and dilution through the inability of bound caffeine to diffuse cancel each other out[72],
- apparent metabolite concentrations are proportional to the sweat volume (see Supplementary Fig. 8, Eq. (1)), and finally,
- sweat volumes are time dependent, but the same for all metabolites at one time-point.

A mathematical formulation of the problem of fluctuating sweat volumes is given in Eq. (1), where $\widetilde{M}(t)$ is the measured mass vector of the internal metabolites and

$C(t)$ is the underlying concentration vector. $V_{sweat}(t)$ is a time-dependent volume that represents the sampled sweat volume. The resulting mathematical model is explained in detail in the Supplementary Note 3: Mathematical Model. Briefly, we describe the kinetics of caffeine metabolism with a system of ordinary differential equations (Supplementary Information, Supplementary Note 3: Mathematical Model, Eq. (2)). Subsequently we connect the solution of this equation over the sweat volume to the concentrations measured in the caffeine capsule study. Our model only contains variables that are either known and are thus fixed (volume of distribution, bioavailability, and ingested dose of caffeine) or have a concrete physical meaning but are unknown and need to be fitted (kinetic parameters, initial concentrations of paraxanthine, theobromine, and theophylline, sweat volumes). It allows to estimate absolute concentrations of tri- and dimethylxanthines in the finger sweat. Note that $V_{sweat}(t)$ is not constant over time and unknown and thus a unique fitting parameter at each sampled time-point. Therefore, the number of parameters that need to be fitted for the model is equal to the number of time-points (one $V_{sweat}$ value per time-point) plus the number of parameters of the kinetic model. This requires the simultaneous fitting of the kinetics of multiple metabolites upon assuming that at each time-point $V_{sweat}(t)$ is constant for similar metabolites (Eq. (2)). By doing so the amount of data points that can be used for fitting is multiplied by the number of metabolites while the number of parameters for $V_{sweat}(t)$ stays constant. Thus (as long as the kinetic model is not overly complex) the system is sufficiently determined and data fitting is feasible.

$$\widetilde{M}(t) = V_{sweat}(t)\,C(t) \qquad (1)$$

$$V_{sweat} = V_{sweat}^{caffeine} = V_{sweat}^{paraxanthine} = V_{sweat}^{theobromine} = V_{sweat}^{theophylline} \qquad (2)$$

Caffeine and its major catabolic products paraxanthine, theobromine and theophylline were modelled subject to the following constraints: first order kinetics for all reactions ($k_1$ to $k_8$) with $0 \le k_1 \le 10\,h^{-1}$ and $0 \le k_{2-8} \le 0.2\,h^{-1}$; initial concentration of 0 for caffeine and $0 \le C_0^i \le 1\,\mu g\,L^{-1}$ for dimethylxanthines; and variability of $V_{sweat}$ between $0.05 \le V_{sweat}(t) \le 4\,\mu L$. Generally, literature values of kinetic constants and sweat rates (without exercise) are well within the bounds of the model[40,51,73,74]. Finally, Supplementary Eq. (15) was used to fit the experimental data of all volunteers of the caffeine capsule study normalised by the machine standard individually. Fitting was performed in Python 3.7 with the SciPy package (version 1.6.1) using the curve_fit function and the integrated trust region reflective algorithm with default numerical tolerances ($10^{-8}$)[75]. To find optimal settings for the fitting procedure we performed a systematic investigation of the hyperparameters in the Supplementary Note 4: Sensitivity Analysis. There, our implementation of the generalised, adaptive robust-loss function[76] in combination with Monte Carlo sampling of initial parameters for 100 times and selecting the solution with the lowest overall loss resulted in the smallest errors. Therefore, the same settings were adopted for this study. Moreover, with the estimated CVs associated to the fitting procedure from the Sensitivity Analysis we calculated confidence intervals (n = 120, df = 93), which are shown as error bars in Fig. 5b, c, e, and f (and Supplementary Table 2). Finally, we performed a PCA of the standard scaled kinetic constants of caffeine degradation ($k_2$, $k_3$, $k_4$, $k_5$) of all volunteer profiles (Fig. 5d).

**Programs for mathematical modelling**. PCA of kinetic parameters (Fig. 5d) was performed with Python 3.7 and scikit-learn (version 0.23.2). Levene-test in sensitivity analysis was performed with Python 3.7 and scipy (version 1.6.1). The mathematical modelling and sensitivity analysis was performed with Python 3.7 heavily relying on packages scipy (version 1.6.1) and robust-loss-pytorch (version 0.0.2).

**Reporting summary**. Further information on research design is available in the Nature Research Reporting Summary linked to this article.

## Data availability

The data supporting the findings from this this study are available within the manuscript and its supplementary information. The metabolomics datasets have been deposited in the MetaboLights repository with the accession numbers MTBLS2772 and MTBLS2776[77]. Any remaining raw data will be available from the corresponding author upon reasonable request. Source data are provided with this paper.

## Code availability

The code for mathematical model fitting and sensitivity analysis is available on GitHub (https://github.com/gotsmy/finger_sweat and https://doi.org/10.5281/zenodo.5222967)[78].

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

## Acknowledgements

We acknowledge support by the Mass Spectrometry Centre of the Faculty of Chemistry, University of Vienna and the Joint Metabolome Facility, University of Vienna and Medical University of Vienna. Both facilities are members of the Vienna Life Science Instruments (VLSI). We would also like to thank the Hochschuljubiläumsstiftung (HJS) for their financial support during the course of this research project. We are grateful to Lukas Skos for help with HepG2 experiments. Figure 1 was created with the help of Servier Medical Art (https://smart.servier.com) and the schematic of the Vanquish UPLC and Q Exactive HF has been created with BioRender (www.BioRender.com).

## Author contributions

J.B. performed research, interpreted data, analysed data and wrote the manuscript, M.G. performed research, analysed and interpreted data, L.N. performed research and analysed data, A.B. interpreted data and wrote the manuscript, B.N. performed research, A.S. performed research, L.J. performed research and analysed data, M.L.F. performed research and analysed data, C.L. performed research and analysed data, J.Z. analysed data and interpreted data, S.M.M. performed research, analysed, interpreted data and wrote the manuscript, C.G. conceptualised the project, interpreted data and wrote the manuscript.

## Competing interests

The authors declare no competing interests.

### Ethics approval

The study protocol was performed in accordance with the University of Vienna and has been approved by the ethical committee of the University of Vienna (reference number 00337). Written informed consent has been obtained from all volunteers participating in this study. The informed consent covers an information sheet about the purpose of the study, a questionnaire for minimal personal information and certificate of consent.
