## [Peer Review File · Nature Communications]

REVIEWER COMMENTS

Reviewer #1 (Remarks to the Author):

In this study, an optimized workflow for the analysis of sweat from the fingertips was reported. Proof-of-principle studies based on the consumption of coffee or caffeine capsules involving a high sampling rate provided evidence of the feasibility of the approach. The study provides a mode for biomonitoring metabolites in humans.

Major comments

1. Metabolomics is a science mainly on endogenous metabolites, the metabolites of coffee are not endogenous. In this manuscript, authors didn't distinguish them. In fact the work in this manuscript had no relationship with metabolomics, it is a typical xenobiotic monitoring in sweat.
2. Sweat from the fingertips is very useful for biomonitoring the metabolites of a xenobiotic, but it is not easy to quantify them. Especially, fingers are easily polluted, washing without soap is not easy to remove the pollutants in daily life. In the meantime, the absolute concentrations of metabolites are influenced by the water in the body, exercise, and environment (temperature and humidity etc.).
3. Authors should show the stability of metabolite concentrations at different times of sampling, drinking water, diet (fasting or not) etc.

Minor comments

1. In Figure 1 (A), the graph of sampling step should be modified based on the description in the section "Collection of Sweat from the Fingertip", the sampling unit was placed between thumb and index finger.
2. Please specify the model or product name of the filter paper used in the experiments.
3. The hands of volunteers were cleaned by tap water and dried with disposable paper towels, however, only unused sampling units were extracted as blank sample, the potential contaminants from tap water and disposable paper towels also should be taken into account. The blank should be considered.
4. Line 119 author said "delivered more than 5000 reproducible sweat-specific features", I think this part should be deleted or a literature is cited.
5. Line 129, "Principle component analysis" should be "Principal component analysis".
6. linear ranges and LOD should be expressed with the concentration unit.

7. Line 333-334: “targeted assays based on the analysis of sweat from the fingertips can be successfully established directly from metabolomic discovery studies”, this sentence is not right, because caffeine is not an endogenous metabolite. Authors should modify the related description in the manuscript.

Reviewer #2 (Remarks to the Author):

This is a very interesting manuscript which has already been pre-published on bioRxiv (<https://doi.org/10.1101/2020.11.06.369355>) considers a simple sampling technique for obtaining non-invasive sweat samples that are then analysed with High-resolution MS using a Q Exactive HF orbitrap hyphenated with an ultra118 high-performance liquid chromatography (UHPLC) system. It is claimed that this ‘delivered more than 50,000 reproducible sweat-specific features per analysis’. Preliminary results suggest it is possible to track time dependent variations in administered xenobiotics like caffeine in samples taken from volunteers.

Overall the work is impressive, and the sampling approach is indeed simple and non-invasive. However, the paper is more suited for a mainstream analytical journal, like Analytical Chemistry, ACA, Talanta, The Analyst rather than Nature Communications. I say this as the analytical approach overall needs to be rigorously reviewed within that community, which is its natural home in the literature, where similar papers continue to appear; for example, “Dry sweat as sample for metabolomics analysis, M.M. Delgado-Povedano, L.S. Castillo-Peinado, M. Calderón-Santiago, M.D. Luque de Castro, F. Priego-Capote, Talanta, 208, 2020; <https://doi.org/10.1016/j.talanta.2019.120428>”, the latest of a series of papers on this topic from the Luque de Castro group (not cited in this paper).

When a manuscript claims 50,000 components can be found in a sample, it arises from the exquisite sensitivity and selectivity of the analytical technique, which clearly so in this case. When this happens, the potential for sampling dependent error quickly raises its head, and attention then is focused on whether the sample audit trail is secure enough, which is debatable in this case. Lots of questions can be asked about the security of the sampling approach, e.g.;

- Were the fingertips washed with any solvent to remove any residual soap from showering, residue from touching surfaces etc prior to handwashing for the trail?

- For the sweat collection the sampling unit was made from filter paper 1cm² diameter using a puncher was there any pre-treatment of the filter paper prior to use as the sample unit?
- Did the pressure of the sampling unit being held between the thumb and the finger effect the sweating rate?
- Was there a comparison done on without pre-wetting??
- How was the sweating rate determined if the sample was pre-wetted?
- Is 1 minute long enough to capture sweat each individual has a different sweating rate?

Care must also be exercised when dismissing other sample matrices, as being unsuitable for 'frequent sampling' of critical biomarkers, when we are witnessing a revolution in the personalised care of diabetes due to wearable patches with integrated sensors that allow continuous monitoring (line 57). It could be said in response that the proposed approach may allow samples to be obtained relatively easily, but it requires hugely sophisticated analytical equipment and experienced trained personnel to generate the results. If this approach is suitable for this purpose, why are the individual caffeine profiles restricted to a very small series of points? Furthermore, the modelling is interesting as it does allow descriptive model parameters to be estimated, but the story gets clear when these are used to modify data. I have many more questions related to the analytical approach, sample integrity, comparisons with induced and natural sweating, relationship between sampling approach and the volunteer profile (some people do not sweat significantly through the fingertip for a variety of reasons), number of data points in the time series (profiles need to be more convincing), and a larger cohort of volunteers to explore the optimum sampling approach more convincingly.

These criticisms are not to take away from the effort the team has made to generate the data they have thus far. But rather it is to emphasise the importance of placing this paper in its natural home in the analytical literature, where these questions can be rigorously reviewed and debated.

Reviewer #3 (Remarks to the Author):

This study proposes a method for biomonitoring using sweat from the fingertip. The following comments address the mathematical modelling. Indeed, the model is not presented in totally clear details. Specific comments are reported below.

1. There is no information about the quality of the parameters estimation. As an example, one may report the confidence intervals for each of the estimated parameters. This is important to judge whether the model is sufficiently robust.
2. Some more details should be reported about the used approach for the parameters estimation (algorithm, required tolerance in the minimization of the cost function, etc.).
3. In addition, it should also be reported how the adjusted Rsquare varies for the different 100 runs of the Monte Carlo method for the initial parameters value. Indeed, if there is large variation in the Rsquare for the different initial values, this may be an indication that the numerical approach is not the best one.
4. It appears that Table S2 reports the estimated parameters (incidentally, I suggest “estimated” rather than “recovered”). However, from the model description it appears that you are not estimating $Q_{\text{free_caf}}(t)$, but $SV(t)$, though $SV(t)$ is then used to compute $Q_{\text{free_caf}}(t)$ (the fitted variable) according to eq. at page 24. Thus, I would report $SV(t)$ in Table S2, rather than $Q_{\text{free_caf}}(t)$.
5. Further comment about the eq. at page 24 relating $Q_{\text{free_caf}}(t)$ and $SV(t)$ is for Bioavailability and Vdistribution: I guess you fixed them to reasonable values, but unless I have missed, it seems such values were not reported.
6. It is unclear to me what may be the reader’s convenience to read the solution of the eq. S2-S5: I think eq. S6-S9 may be omitted. At contrast, eq. S10 is useful, since the parameter is then used subsequently (for instance in Table S3). However, it should be somehow “introduced” here, since as it is now one may not understand why it is reported (one looks to the equations above, and does not see such parameter, thus apparently there’s no need to report it: it has to be explained that it is used later).
7. In $C_{\text{free_caf}}$ expression following eq. S9, please specify $C_{\text{free_caf}}(0)$ for more clarity.
8. The explanation for assuming $k_6=k_8$ is not totally clear to me (“since they have the biggest influence on the time of maximum concentration”): I would better understand just something like “since the two metabolites were reported to have similar kinetics”.

9. What is the rationale for Q_{free_caf} range between 10 and 1000 nAUC? It appears it was not explained/justified.

10. As regards Bioavailability and $V_{distribution}$, it is unclear what you mean saying that they are analogous to SV. At contrast, I agree about the fact that they cannot be estimated unambiguously in your model formulation.

11. Few lines below, it is correct that the kinetic parameters can be estimated, but to me it is unclear the sentence " $Q_{free_caf}(t)$ scales the fitted curve on the y-axis, whereas its shape - and thus the kinetic parameters that describe individual metabolic differences" (and just in the line below I would replace "constants" with "parameters").

12. In addition, another sentence unclear is "Note that since $Q_{free_caf}(t)$ is not constant over time and a fitting parameter at each instance" (I guess maybe a word is missing); also, in line below I would suggest "at least equal", and few lines below I would keep on speaking of "kinetics" rather than "dynamics".

13. "By doing so the number of data points that can be fitted is multiplied": multiplied by what? Please rephrase.

Reviewer #4 (Remarks to the Author):

This is a smart and well performed study, describing a very well thought and clearly non-invasive methodology to investigate absorption and metabolism of coffee phytochemicals.

Despite the clear "untargeted" approach, the work ends up to be indeed targeting the main components, which makes sense. However, considering this focus, a little more attention on the real known metabolites of certain coffee components should have been paid. If the methyl-xanthines have been dealt with according to the literature, the same could not be said for phenolic compounds. Actually, the main hydroxycinnamic acids present in coffee are known to be vastly modified after ingestion along the GI tract and post-absorption, with deconjugation from the quinic

acid moisture and subsequent phase II metabolism at hepatic level. As a reference, we could cite Nat. Prod. Rep., 2017, 34, 1391, that clearly reports that caffeic and ferulic acid and their phase II conjugates vastly overcome chlorogenic acids in circulation. Additionally, the gut microbiota has also been described to modify the structure of the unabsorbed phenolics, giving origin to dihydrocaffeic and dihydroferulic acids, that are subsequently absorbed and further conjugated in the liver to circulate then at systemic level. The authors should have considered these chemical species in their search for sweat metabolites, instead of the simple intact 5-CQA. Moreover, a more frequent sampling also for phenolic metabolites could have helped unravelling possible variations among subjects. For this reason, I find figure 3C very misleading, as it leaves the impression that we should expect to find exactly the same compounds that are present in coffee in our sample of sweat, where this is clearly not the case.

The sentence in the abstract "Moreover, the determined limits of detection demonstrate that the ingestion of 200 µg of

a xenobiotic may be sufficient for its detection in sweat from the fingertip", originating from the reasoning present at page 7, is a little too strong. The volume available for dilution in the body depends on the capacity of a metabolite to enter cells and the bioavailability of several xenobiotics is quite low. I would not dare making such a statement.

Were N-methylpyridinium and N-methylnicotinamide measured and followed, as they have recently been reported to be good markers of coffee intake.

Very recently, also phenylalanine, tyrosine, energy metabolism, steroid hormone biosynthesis, and arginine biosynthesis and metabolism have been reported to be modulated by coffee intake. Was this somehow confirmed by your experience?

Response to Reviewers

Reviewer #1 (Remarks to the Author):

Major comments

1. Metabolomics is a science mainly on endogenous metabolites, the metabolites of coffee are not endogenous. In this manuscript, authors didn't distinguish them. In fact, the work in this manuscript had no relationship with metabolomics, it is a typical xenobiotic monitoring in sweat.

Answer: The present study focusses on the analysis of exogenous metabolites from coffee extracts and caffeine from a capsule, the hepatic metabolic products of caffeine, as well as endogenous metabolites (e.g. dopamine and adenosine amongst others, Figure 1 and Supplementary Table 6) in sweat from the fingertips. Thus, our study includes the analysis of both exogenous and endogenous metabolites by an untargeted mass spectrometry-based metabolomics approach. The term “*metabolomics*” refers to the analysis of small molecule metabolites in cells, tissues and body fluids. Some define the metabolome to include exclusively endogenous metabolites (Holmes et al., 2008, Cell, DOI: 10.1016/j.cell.2008.08.026; Guijas et al., 2018, Nat. Biotechnol., DOI: 10.1038/nbt.4101) and differentiate it from “*metabonomics*”, the latter extending the metabolic response to drugs and environmental changes (Assfalg et al., 2008, PNAS, DOI: 10.1073/pnas.0705685105). The consortium around the human metabolome database (HMDB) defines the human metabolome more broadly as the complete collection of small molecules including endogenous and exogenous metabolites that humans ingest, metabolize or catabolize (Wishart et al., 2018, Nucl. Acid. Res., DOI: 10.193/nar/gkx1089). In order to avoid misinterpretation as suggested by the reviewer, we now opted to concisely address this issue in the introduction and refer strictly to “*metabolic phenotyping*” or “*metabolic profiling*” in our study. The metabolic phenotyping is defined as the assessment of the characteristic metabolite profile from biofluids or tissue samples (Nicholson et al., 2012, Nature, DOI: 10.1038/nature11708) reflecting host genome and its interaction with environmental factors, including diet etc. (Holmes et al., 2008, Cell, DOI: 10.1016/j.cell.2008.08.026).

2. Sweat from the fingertips is very useful for biomonitoring the metabolites of a xenobiotic, but it is not easy to quantify them. Especially, fingers are easily polluted, washing without soap is not easy to remove the pollutants in daily life. In the meantime, the absolute concentrations of metabolites are influenced by the water in the body, exercise, and environment (temperature and humidity etc.).

Answer: Our instrumentation is highly-sensitive to detect contaminants in the fingertip (from the skin or our daily used products like shampoo). In our experience, however, the metabolite contaminants on the fingers are thoroughly removed by the presented sample collection strategy. In fact, we washed the fingers initially with soap, but this introduced a stronger bias due to the surfactants contained in soap. Eccrine sweat is mainly comprised of water (>99%). As the presented work-up involves aqueous conditions, this procedure selects for relatively hydrophilic metabolites and does not efficiently enrich lipidic metabolites. Thus, the focus on relatively hydrophilic metabolites justifies washing without soap.

The reviewer is correct by stating that the absolute concentrations of metabolites in fingertips are influenced by many uncontrollable factors. In this context, the beneficial effects of refraining from specific dietary products to obtain an optimal metabolite baseline in individuals may be emphasized. Also, repeated sampling of control groups revealed only low fluctuations of the assessed analytes over time (Supplementary Figure S2). However, absolute quantification of metabolites in the fingertips is indeed challenging due to the difficulty to control for sweat volume (i.e. the rate of sweat formation). This has impeded the use of sweat from the fingertips as a meaningful biofluid for metabolic phenotyping so far. Thus, we developed a useful strategy to circumvent the need for absolute quantitative measurements by mathematically modelling of the kinetic profile of biochemically related pairs of molecules such as caffeine and its primary metabolic degradation products. This allowed us to normalize individual rates of sweat formation and results in absolute quantitative metabolomics.

3. Authors should show the stability of metabolite concentrations at different times of sampling, drinking water, diet (fasting or not) etc.

Answer: We thank the reviewer for this suggestion. As stated above (Answer to point 2), repeated sampling of control groups revealed only low fluctuations of the assessed analytes over time (Supplementary Figure 2). Additionally, we have addressed this aspect in a principal component analysis (Figure 5d) showing that the metabolite profiles of individuals cluster over a two-year time-span despite different dietary behavior (including drinking). Thus, hepatic caffeine metabolism as assessed from sweat of the fingertips is largely stable for each individual independent of time.

Minor comments

1. In Figure 1 (A), the graph of sampling step should be modified based on the description in the section "Collection of Sweat from the Fingertip", the sampling unit was placed between thumb and index finger.

Answer: We thank the reviewer for this suggestion. The figure was adapted to show the actual sampling technique.

2. Please specify the model or product name of the filter paper used in the experiments.

Answer: We have now included the detail about the product and the company (precision wipes, number 7552, 11x21 cm from Kimtech Science, Kimberly-Clark Professional, USA).

3. The hands of volunteers were cleaned by tap water and dried with disposable paper towels, however, only unused sampling units were extracted as blank sample, the potential contaminants from tap water and disposable paper towels also should be taken into account. The blank should be considered.

Answer: To rule out contamination stemming from the filter blank, tap water or paper towels we have now analyzed 10 samples from each blank type. We did not detect any of the discussed metabolites in these blanks (see results and source data).

4. Line 119 author said "delivered more than 50'000 reproducible sweat-specific features", I think this part should be deleted or a literature is cited.

Answer: We have moved this statement to the methods section. The number was not intended as a figure of merit, but as a benchmark of the experimental conditions and data acquisition typical for mass spectrometry-based metabolic profiling.

5. Line 129, "Principle component analysis" should be "Principal component analysis".

Answer: We have now corrected the typo.

6. linear ranges and LOD should be expressed with the concentration unit.

Answer: Linear ranges and LOD are now expressed with the concentration unit.

7. Line 333-334: "targeted assays based on the analysis of sweat from the fingertips can be successfully established directly from metabolomic discovery studies", this sentence is not right, because caffeine is not an endogenous metabolite. Authors should modify the related description in the manuscript.

Answer: Since this section discusses the exogenous metabolite caffeine and the hepatic metabolic products theophylline, theobromine and paraxanthine we adapted the sentence to include metabolic phenotyping according to the discussion under Major Comments, Point 1, as follows: "This suggests

that targeted assays based on the analysis of sweat from the fingertips can be successfully established directly from metabolic phenotyping”.

Reviewer #2 (Remarks to the Author):

1. Overall the work is impressive, and the sampling approach is indeed simple and non-invasive. However, the paper is more suited for a mainstream analytical journal, like Analytical Chemistry, ACA, Talanta, The Analyst rather than Nature Communications. I say this as the analytical approach overall needs to be rigorously reviewed within that community, which is its natural home in the literature, where similar papers continue to appear; for example, “Dry sweat as sample for metabolomics analysis, M.M. Delgado-Povedano, L.S. Castillo-Peinado, M. Calderón-Santiago, M.D. Luque de Castro, F. Priego-Capote, Talanta, 208, 2020; <https://doi.org/10.1016/j.talanta.2019.120428>”, the latest of a series of papers on this topic from the Luque de Castro group (not cited in this paper).

Answer: We thank the reviewer for the kind remarks on our work. The main pillars of this manuscript are the analytical method validation, the mathematical modeling to account for the rate of sweat formation and the cohort studies with relevance to predictive, preventive and personalized medicine (PPPM). As these are tightly connected aspects of the main research question, they clearly underline the interdisciplinary nature of this approach and are thus expected to be interesting to a broad readership from analytical chemistry to clinical diagnostics, including PPPM. Ongoing research will establish the relation of individual molecular patterns obtained by analyzing sweat from the fingertips with respect to disease states, risk for disease or the validation of therapeutic strategies. Moreover, we have now also included a cell culture model, which demonstrates hepatic metabolism of caffeine, and release of primary metabolites into the supernatants that support our findings from the analysis of sweat from the fingertips after ingestion of caffeine.

We have also included the paper suggestion, which demonstrates the potential of sweat as matrix for biomonitoring in humans.

2. When a manuscript claims 50,000 components can be found in a sample, it arises from the exquisite sensitivity and selectivity of the analytical technique, which clearly so in this case. When this happens, the potential for sampling dependent error quickly raises its head, and attention then is focused on whether the sample audit trail is secure enough, which is debatable in this case. Lots of questions can be asked about the security of the sampling approach, e.g.:

2.1. Were the fingertips washed with any solvent to remove any residual soap from showering, residue from touching surfaces etc. prior to handwashing for the trail?

Answer: This question was similarly raised by Reviewer 1, Major comments, Point 2. In our approach, the hands (incl. fingertips) were thoroughly washed with warm tap water and cleaned with disposable paper towels before each sampling time-point. After washing, donors were not allowed to touch any surfaces. In our experience, the metabolite contaminants on the fingers are thoroughly removed by the presented sample collection strategy. In fact, we washed the hands initially with soap, but this introduced a stronger bias due to the surfactants contained in soap. Eccrine sweat is mainly composed of water (>99%). As the presented work-up involves aqueous conditions, this procedure selects for relatively hydrophilic metabolites and does not efficiently enrich lipidic metabolites. Thus, the focus on relatively hydrophilic metabolites justifies washing without soap.

It may be noted that skin contaminants can be detected and confidently identified. For example, we found tangeritin (a flavone of citrus peels) as a contaminant in sweat of the fingertip of a volunteer, after having peeled an orange. This underlines the necessity to strictly adhere to the sample collection protocol. We have additionally included a time series of two control volunteers that showed only small fluctuations in the assessed metabolites over time using this method (Supplementary Figure 2).

Extracted ion chromatogram of tangeritin found in sweat.

2.2. For the sweat collection the sampling unit was made from filter paper 1cm² diameter using a puncher was there any pre-treatment of the filter paper prior to use as the sample unit?

Answer: We used Kimtech precision wipes as sampling units. All steps were performed wearing gloves. Utensils were cleaned with 70% ethanol prior to use. The wipes were folded and put in clean plastic bags. Filter papers were stamped out within the plastic bag, which was then discarded and the obtained sampling units were transferred into a clean plastic bag with tweezers that have been cleaned with 70% ethanol prior to use. The sampling units were pre-wetted with 3 μ L water and without further pre-treatment. We have additionally extracted and analyzed >10 blank sample units during the course of our studies and found negligible background levels (see results and source data).

2.3. Did the pressure of the sampling unit being held between the thumb and the finger effect the sweating rate? And: How was the sweating rate determined if the sample was pre-wetted?

Answer: Indeed, it is challenging to determine the rate of sweat formation on the fingertips and the associated sweat volume per unit time. The same reason also largely impeded the absolute quantification of metabolites detected in sweat from the fingertip. We solved this problem by mathematically modelling the kinetic profile of biochemically related pairs of molecules such as caffeine and its hepatic primary metabolites (theobromine, theophylline, paraxanthine). The applied dynamic metabolic network modelling allowed us to normalize the data and estimate the rate of sweat formation at each sampling time point. Consequently, our sampling approach is independent of the actual rate of sweat formation and thus, independent of factors that influence the rate of sweat formation, such as pressure, temperature etc. The feasibility of such an approach is an additional key finding of our work.

2.4. Was there a comparison done on without pre-wetting??

Answer: Yes, we have investigated different volumes of pre-wetting and also dry filter papers. Metabolite concentrations found in pre-wetted filter papers were significantly higher compared to dry filter papers. Thus, our data is in accordance with previously published findings (Kuwayama et. al., 2012, Anal. Bioanal. Chem., DOI: 10.1007/s00216-012-6569-3).

2.5. Is 1 minute long enough to capture sweat each individual has a different sweating rate?

Answer: Yes, the small amounts of sweat were sufficient to analyze also challenging metabolites. Actually, we collected the sweat from a total of 2 minutes. After washing and drying the hands, the procedure includes a 1 min waiting time after which the sampling between the thumb and index finger is carried out for 1 min. The mathematically determined boundaries for the estimated sweat volumes are between 0.05–4 μ L (Figure 5a).

2.6. Care must also be exercised when dismissing other sample matrices, as being unsuitable for 'frequent sampling' of critical biomarkers, when we are witnessing a revolution in the personalized care of diabetes due to wearable patches with integrated sensors that allow continuous monitoring (line 57). It could be said in response that the proposed approach may allow samples to be obtained relatively

easily, but it requires hugely sophisticated analytical equipment and experienced trained personnel to generate the results.

Answer: We thank the reviewer for this remark, we have now included limitations of our approach in the discussion.

2.7. If this approach is suitable for this purpose, why are the individual caffeine profiles restricted to a very small series of points?

Answer: In order to improve the mathematical modelling, we have expanded the cohort and increased the number of sampling time points (s. description of cohort C.2). Each volunteer was sampled 20-times within 24 h with at least one sampling time point per hour during the first 14 h. This was feasible and maintained the quality of the results. Of note, the sampling time-points between 8 and 14 h were crucial to consolidate the mathematical model. It can thus be argued that the number of time points can be increased without affecting the quality of the results, which may enable longitudinal studies.

2.8. Furthermore, the modelling is interesting as it does allow descriptive model parameters to be estimated, but the story gets clear when these are used to modify data. I have many more questions related to the analytical approach, sample integrity, comparisons with induced and natural sweating, relationship between sampling approach and the volunteer profile (some people do not sweat significantly through the fingertip for a variety of reasons), number of data points in the time series (profiles need to be more convincing), and a larger cohort of volunteers to explore the optimum sampling approach more convincingly.

Answer: As mentioned under point 2.7., we have increased the cohort from 15 to 47 volunteers and increasing the number of sampling time-points in the adapted cohort C.2 (s. methods section). The enlarged cohort consolidated the mathematical model and confirmed the main findings of cohort C.1. The enlarged cohort is described in the methods section and the findings were implemented into the results and discussion. Overall, this manuscript presents data stemming from the analysis of 1'792 sweat samples. For a collaboration study, the samples were internationally shipped by postal service, which did not lead to obvious bias in the sample composition (Lin *et.al.* 2021, *Cereb. Cortex*, DOI: 10.1093/cercor/bhab005). Moreover, we now provide an extensive description of the mathematical validation of our approach including an assessment of its (numerical) robustness. The collective data underlines the robustness of the approach, which will be investigated in further detail to address the impact of the mentioned points.

Reviewer #3 (Remarks to the Author):

We would like to thank the reviewer for the careful evaluation of our manuscript, which we could use to significantly improve its quality. Below we address these comments individually. Light blue text in quotes and italics indicates additions to the manuscript.

A. This study proposes a method for biomonitoring using sweat from the fingertip. The following comments address the mathematical modelling. Indeed, the model is not presented in totally clear details. Specific comments are reported below.

Answer: We agree with the reviewer that due to the brevity of the presentation some aspects of our mathematical model may not have been clear in the original submission. Therefore, we thoroughly revised the main text and now provide extensive documentation in the supplementary material. *Supplementary Notes: Mathematical Model* describes the set-up of the model. *Supplementary Notes: Sensitivity Analysis* describes a systematic analysis of the fitting procedure and the associated errors. Moreover, we now provide a Jupyter Notebook of our implementation to make the model fully transparent and reproducible (https://github.com/Gotsmy/finger_sweat).

B. There is no information about the quality of the parameters estimation. As an example, one may report the confidence intervals for each of the estimated parameters. This is important to judge whether the model is sufficiently robust.

Answer: Thank you for your suggestion. In the current version of our manuscript we address the method's robustness twofold:

In *Supplementary Notes: Sensitivity Analysis* we ran simulations with synthetic data. We evaluated the performance of the fitting procedure and estimated coefficients of variation which we used to calculate the confidence intervals shown in Figure 5b, c, e, f and Supplementary Table 2.

The method's robustness is further corroborated in Figure 5d which shows that kinetic parameters derived multiple times from the same individuals cluster closely together— even if measurements are 1.5 years apart.

1. Some more details should be reported about the used approach for the parameter's estimation (algorithm, required tolerance in the minimization of the cost function, etc.).

Answer: We agree that these data are crucial information. We added following text to the methods section of the main text:

“Fitting was performed with the SciPy Python package using the curve-fit function and a trust region reflective algorithm with default numerical tolerances ($1E-8$) (Virtanen et al. 2020). To find optimal settings for the fitting procedure we performed a systematic investigation of the hyperparameters in the Supplementary Notes: Sensitivity Analysis. There, our implementation of the generalized, adaptive robust loss function (Barron et al., 2019) in combination with Monte Carlo sampling of initial parameters for 100 times and selecting the solution with the lowest overall loss resulted in the smallest errors. Therefore, the same settings were adopted for this study.”

2. In addition, it should also be reported how the adjusted Rsquare varies for the different 100 runs of the Monte Carlo method for the initial parameters value. Indeed, if there is large variation in the Rsquare for the different initial values, this may be an indication that the numerical approach is not the best one.

Answer: We acknowledge the reviewer's concern about Monte Carlo sampling. We investigated the size of error associated to the fitting procedure with different numbers of Monte Carlo replicates in the *Supplementary Notes: Sensitivity Analysis*. We found an advantage of using 100 MC replicates over not using them as described in the Methods Section (see above). Only when using 100 MC replicates the average median relative error of the fit was below 5% (Supplementary Fig. 15).

3. It appears that Table S2 reports the estimated parameters (incidentally, I suggest “estimated” rather than “recovered”). However, from the model description it appears that you are not estimating $Q_{free_caf}(t)$, but $SV(t)$, though $SV(t)$ is then used to compute $Q_{free_caf}(t)$ (the fitted variable) according to eq. at page 24. Thus, I would report $SV(t)$ in Table S2, rather than $Q_{free_caf}(t)$.

Answer: In the original manuscript, we were estimating $Q_{free_caf}(t)$. However, during the revision process, we were able to further improve our model. This now allows us to report SV values (which we renamed V_{sweat} to make them compliant with npj styles). Moreover, we completely removed the - now unnecessary - definition of $Q_{free_caf}(t)$ from the manuscript and changed the caption of Table S2 accordingly as follows:

“Supplementary Table 2: Estimated parameters. Kinetic parameters [h^{-1}], initial (unit-less) concentrations and sweat volumes (V_{sweat}) [μL] fitted by the kinetic model (Figure 5a) with their respective 95 % confidence intervals (calculated from CVs from Supplementary Table 14, $n = 120$, $df = 93$) for donor 1 and donor 2. [...] As comparison the kinetic parameters found in blood plasma after uptake of 5 mg caffeine per kg specimen body mass are listed (Bonati, 1982).”

4. Further comment about the eq. at page 24 relating $Q_{\text{free_caf}}(t)$ and $SV(t)$ is for Bio-availability and $V_{\text{distribution}}$: I guess you fixed them to reasonable values, but unless I have missed, it seems such values were not reported.

Answer: We thank the reviewer for the suggestion. In the original manuscript we were actually using the whole expression of $Q_{\text{free_caf}}(t)$ as fitting parameter and did therefore not need to fix the other parameters to literature values. However, with our revised model we are fixing the bioavailability and the volume of distribution to literature values listed in the *Supplementary Notes: Mathematical Model*, Supplementary Table 6.

5. It is unclear to me what may be the reader's convenience to read the solution of the eq. S2-S5: I think eq. S6-S9 may be omitted. At contrast, eq. S10 is useful, since the parameter is then used subsequently (for instance in Table S3). However, it should be somehow "introduced" here, since as it is now one may not understand why it is reported (one looks to the equations above, and does not see such parameter, thus apparently there's no need to report it: it has to be explained that it is used later).

Answer: We thank you for your feedback on the equations. Indeed, the supplementary equations section was unclear in the original manuscript. We agree that the solutions to the differential equations are no novelty. Yet, readers might not have an intuition what the solutions will look like. In order to be self-contained, we decided to leave them in the supplementary material, but we removed references to them from the main manuscript.

6. In $C_{\text{free_caf}}$ expression following eq. S9, please specify $C_{\text{free_caf}}(0)$ for more clarity.

Answer: Thank you for pointing at that inconsistency in the notation. Originally $C_{\text{free_caf}}$ was in fact supposed to mean $C_{\text{free_caf}}(0)$. As mentioned above we extensively rewrote the supplementary material and improved the consistency of our notation. $C_{\text{free_caf}}(0)$ (which we renamed $C_{\text{free_caf}}$) is now defined in Supplementary Equation 4 (in *Supplementary Notes: Mathematical Model*).

7. The explanation for assuming $k_6=k_8$ is not totally clear to me ("since they have the biggest influence on the time of maximum concentration"): I would better understand just something like "since the two metabolites were reported to have similar kinetics".

Answer: We agree that the reviewer's suggestion is a better way of saying it. We completely removed this sentence, as—after revising our model—we dropped that assumption and now fit both k_6 and k_8 using tighter bounds on the kinetic parameters found recently (Grzegorzewski et al., 2021, *Nucl. Acid. Res.*, DOI: 10.1093/nar/gkaa990).

8. What is the rationale for $Q_{\text{free_caf}}$ range between 10 and 1000 nAUC? It appears it was not explained/justified.

Answer: We agree that this was a valid concern in our old model. However – as explained above – with the revised model the term that is fitted boils down from $Q_{\text{free_caf}}$ to V_{sweat} . We calculated the expected range of sweat volume from literature sweat fluxes as described in the *Supplementary Notes: Sensitivity analysis* (see shaded panel below). Sweat volumes we expected from literature values are well within the bounds we used for our model ($0.05 \leq V_{\text{sweat}} \leq 4 \mu\text{L}$).

"The sweat volume, V_{sweat} , is calculated from the sweat flux by multiplying by 2 cm^2 (sampling area) and 2 min (1 min sweating and 1 min sampling after washing hands)."

9. As regards Bioavailability and $V_{\text{distribution}}$, it is unclear what you mean saying that they are analogous to SV. At contrast, I agree about the fact that they cannot be estimated unambiguously in your model formulation.

Answer: As we thoroughly revised the kinetic model and its description this issue is now resolved and presented much clearer. In the Methods Section we state:

“Our model only contains variables that are either known and are thus fixed (volume of distribution, bioavailability, and ingested dose of caffeine) or have a concrete physical meaning but are unknown and need to be fitted (kinetic parameters, initial concentrations of paraxanthine, theobromine, and theophylline, sweat volumes).”

10. Few lines below, it is correct that the kinetic parameters can be estimated, but to me it is unclear the sentence “ $Q_{free_caf}(t)$ scales the fitted curve on the y-axis, whereas its shape- and thus the kinetic parameters that describe individual metabolic differences” (and just in the line below I would replace “constants” with “parameters”).

Answer: Thanks for pointing out this inconsistency. In the previous manuscript $Q_{free_caf}(t)$ was used for normalization purposes. In the revised version we improved the consistency of the notation and formulated the mathematical problem in the original variables as well as in normalized variables. Thus, the indicated sentence was omitted.

11. In addition, another sentence unclear is “Note that since $Q_{free_caf}(t)$ is not constant over time and a fitting parameter at each instance” (I guess maybe a word is missing); also, in line below I would suggest “at least equal”, and few lines below I would keep on speaking of “kinetics” rather than “dynamics”.

Answer: We agree that this paragraph was unclear and updated it accordingly. “at least equal” would be appropriate if the sentence stopped before “plus”. However, in its current form we argue that “equal to” is the more precise phrasing as follows:

“Note that $V_{sweat}(t)$ is not constant over time and unknown and thus a unique fitting parameter at each sampled time point. Therefore, the number of parameters that need to be fitted for the model is equal to the number of time points (one V_{sweat} value per time point) plus the number of parameters of the kinetic model. This requires the simultaneous fitting of the kinetics of multiple metabolites upon assuming that at each time point $V_{sweat}(t)$ is constant for similar metabolites (Equation 2).”

12. “By doing so the number of data points that can be fitted is multiplied”: multiplied by what? Please rephrase.

Answer: Thanks for pointing out this unclarity, we updated the sentence accordingly.

“By doing so the amount of data points that can be used for fitting is multiplied by the number of metabolites while the number of parameters for $V_{sweat}(t)$ stays constant, thus (as long as the kinetic model is not overly complex) the system is sufficiently determined and data fitting is feasible.”

Reviewer #4 (Remarks to the Author):

1. Despite the clear "untargeted" approach, the work ends up to be indeed targeting the main components, which makes sense. However, considering this focus, a little more attention on the real known metabolites of certain coffee components should have been paid. If the methyl-xanthines have been dealt with according to the literature, the same could not be said for phenolic compounds. Actually, the main hydroxycinnamic acids present in coffee are known to be vastly modified after ingestion along the GI tract and post-absorption, with deconjugation from the quinic acid moisture and subsequent phase II metabolism at hepatic level. As a reference, we could cite Nat. Prod. Rep., 2017, 34, 1391, that clearly reports that caffeic and ferulic acid and their phase II conjugates vastly overcome chlorogenic acids in circulation. Additionally, the gut microbiota has also been described to modify the structure of

the unabsorbed phenolics, giving origin to dihydrocaffeic and dihydroferulic acids, that are subsequently absorbed and further conjugated in the liver to circulate then at systemic level. The authors should have considered these chemical species in their search for sweat metabolites, instead of the simple intact 5-CQA. Moreover, a more frequent sampling also for phenolic metabolites could have helped unravelling possible variations among subjects. For this reason, I find figure 3C very misleading, as it leaves the impression that we should expect to find exactly the same compounds that are present in coffee in our sample of sweat, where this is clearly not the case.

Answer: We thank the reviewer for this discussion point. We used coffee and subsequently the caffeine capsule in a larger cohort as a proof-of-principle for our biomonitoring approach. For this purpose, we focused here on caffeine and its primary metabolites and not necessarily on other components of coffee extracts because they are the widely accepted bioactive entities that evoke specific biological responses in humans. For example, the biological effect of caffeine in countering sleep pressure is mediated via antagonizing adenosine receptors (Jagannath et al., 2021, Nat. Commun., DOI: 10.1038/s41467-021-22179-z). Indeed, we found adenosine to be significantly up-regulated in all volunteers after ingestion of a 200 mg caffeine capsule. Moreover, the known hepatic metabolic conversion of caffeine into the primary the metabolites theobromine, theophylline and paraxanthine is dependent on cytochrome P450 enzymes. We performed an additional *in vitro* assay using HepG2 cells in which we were able to modulate the conversion of caffeine into specific primary metabolites in the presence or absence of cytochrome P450 enzymes (Supplementary Figure 6), which may partly explain individual differences of caffeine metabolisation.

Monitoring chlorogenic acid and other phenolic compounds from coffee is an interesting point because these xenobiotics are largely transformed by the gut microbiota and would thus allow to draw conclusions about the integrity of this symbiotic relationship. However, we believe that such a study merits a separate publication and goes beyond the scope of this manuscript.

Figure 3C shows the extracted ion chromatogram of chlorogenic acid and its isomers a short 15 min time period after consuming a cup of coffee. We found chlorogenic acid to be rapidly distributed and excreted, on a timescale that does not allow for phase II metabolites to accumulate. In fact, this figure was simply intended to underline the finding that all isomers of chlorogenic acid are equally absorbed, distributed and excreted via sweat from the fingertips. We have now adapted the text to clarify this point.

2. The sentence in the abstract "Moreover, the determined limits of detection demonstrate that the ingestion of 200 µg of a xenobiotic may be sufficient for its detection in sweat from the fingertip", originating from the reasoning present at page 7, is a little too strong. The volume available for dilution in the body depends on the capacity of a metabolite to enter cells and the bioavailability of several xenobiotics is quite low. I would not dare making such a statement.

Answer: This statement was removed from the abstract.

3. Were N-methylpyridinium and N-methylnicotinamide measured and followed, as they have recently been reported to be good markers of coffee intake.

Answer: We have measured N-methylnicotinamide in sweat of participants, however could not correlate it to coffee intake. N-methylpyridinium exceeds the lower mass limit of our mass spectrometric analysis and was not assessed. Interestingly, next to trigonelline, which shows good correlation to coffee consumption (Rothwell et al., 2019, Mol. Nutr. Food Res., DOI: 10.1002/mnfr.201900659), we have found cyclo(leucyl)prolyl significantly upregulated after coffee consumption (not discussed in the text), which is also a known coffee biomarker (Rothwell et al., 2014, Plos One, DOI: 10.1371/journal.pone.0093474).

4. Very recently, also phenylalanine, tyrosine, energy metabolism, steroid hormone biosynthesis, and arginine biosynthesis and metabolism have been reported to be modulated by coffee intake. Was this somehow confirmed by your experience?

Answer: Our study related to coffee consumption were not as detailed as the ones for the caffeine capsule. For example, we studied the coffee consumption with less sampling time points and over 120 mins longitudinally. In this setup, although detected, we did not see changes in the above-mentioned pathways. In contrast, we saw changes in the levels of phenylalanine and tyrosine in most donors in the caffeine capsule study but cannot rule out that it stems from confounding factors, such as nutrition or diurnal effects.

REVIEWERS' COMMENTS

Reviewer #1 (Remarks to the Author):

Authors have addressed all of my issues.

Two minor comments,

1. line3 439-440, "So far, we have verified 250 metabolites with external standards (Supplementary 440 Table 6).", but I can't find the table of 250 metabolites. I found "Supplementary Table 6: List of constants and xed parameters.", and hope authors check it.

2. line 620, "setting the FDR to 0.05 and s0 to 0.1.", it seems some words were missed between "and" and "s0".

Reviewer #2 (Remarks to the Author):

I note the authors response and the efforts they have made to address the reviewers comments. Despite this, my overall comment remains unchanged "Overall the work is impressive, and the sampling approach is indeed simple and non-invasive. However, the paper is more suited for a mainstream analytical journal, like Analytical Chemistry, ACA, Talanta, The Analyst rather than Nature Communications. I say this as the analytical approach overall needs to be rigorously reviewed within that community, which is its natural home in the literature, where similar papers continue to appear..."

Reviewer #3 (Remarks to the Author):

Authors have provided reply to each comment and related changes to the manuscript (plus detailed supplementary information).

Reviewer #4 (Remarks to the Author):

The manuscript is now acceptable.

Response to Reviewers:

Reviewer 1:

1. line3 439-440, "So far, we have verified 250 metabolites with external standards (Supplementary 440 Table 6).", but I can't find the table of 250 metabolites. I found "Supplementary Table 6: List of constants and xed parameters.", and hope authors check it.

We thank the reviewer for this important observation. The Supplementary Table we were referring to was Supplementary Table 17 and part of the excel file called "Supplementary Tables". We have now put both tables from the excel sheet into the Supplementary Information, thus, they are now Supplementary Table 6 and 7 and listed correctly in the manuscript.

2. line 620, "setting the FDR to 0.05 and s0 to 0.1.", it seems some words were missed between "and" and "s0".

We have now explained s0 in more detail.

Reviewer 1-4:

We are very thankful to all the reviewers as their comments really helped in improving our manuscript.